# Global Sensitivity Analysis to Study the Impacts of Bed-Nets, Drug Treatment, and Their Efficacies on a Two-Strain Malaria Model

**Saminu Bala** [1,*,†] and **Bello Gimba** [2,†]

1   Department of Mathematical Sciences, Bayero University, Kano, Nigeria
2   Department of General Studies, School of Health Technology, Jahun, Nigeria; bellogimba@yahoo.co.uk
*   Correspondence: sibala.mth@buk.edu.ng; Tel.: +234-8035-338768
†   These authors contributed equally to this work.

**Abstract:** Malaria is a deadly infectious disease, which is transmitted to humans via the bites of infected female mosquitoes. Antimalarial drug resistance has been identified as one of the characteristics of malaria that complicates control efforts. Typically, the use of insecticide-treated bed-nets (ITNs) and drug treatment are some of the recommended control strategies against malaria. Here, the use of ITNs, drug treatment, and their efficacies and evolution of antimalarial drug resistance are considered to be the major driving forces in the dynamics of malaria transmissions. We formulate a mathematical model of two-strain malaria to assess the impacts of ITNs, drug treatment, and their efficacies on the transmission dynamics of the disease in a human population. We propose a simple mosquito biting rate function that depends on both the proportion of ITN usage and its efficacy. We show that both disease-free and co-existence equilibrium points are globally-asymptotically stable where they exist. The global uncertainty and sensitivity analysis conducted show that if about 95% of malaria cases can be treated with fewer than 5% treatment failure in a population with 95% ITN usage that remains 95% effective, malaria can be controlled. We find that the order in which numerous intervention measures are taken is important.

**Keywords:** intervention; sensitivity; bed-net efficacy; drug resistance; biting rate

**MSC:** 92B05; 97M99

## 1. Introduction

Malaria is one of the most devastating infectious diseases in the world is caused by the protozoan *Plasmodium* and transmitted to humans through the bite of female *Anopheles* mosquitoes. Four species of the parasite, *Plasmodium falciparum*, *Plasmodium vivax*, *Plasmodium ovale*, and *Plasmodium malariae*, infect humans. Since the year 2000, significant progress has been made in fighting malaria. According to the report in [1], between the year 2000 and the year 2015, malaria case incidence was reduced by 41%, and malaria mortality rates were decreased by 62%. The report further indicates that the endemicity of malaria decreased from 108 countries and territories in the year 2000 to 91 in 2016. These changes are attributed to wide-scale deployment of malaria control strategies. These include the use of conventional insecticide-treated bed-nets (ITNs), long-lasting insecticide nets (LLINs), intermittent preventive treatment (IPT) especially for pregnant women during the anti-natal period, reducing mosquito population through the destruction of breeding sites or killing of the larva stage at breading sites that cannot be destroyed, indoor residual spraying (IRS), and the

use of the sterile insect technique [2–8]. Despite the remarkable progress, malaria remains a major cause of mortality and morbidity in many parts of the world. It has a devastating impact on public health and the socio-economic conditions of the people [1,9]. According to the 2015 World Health Organization malaria report [1], it was estimated that 212 million malaria cases occurred globally in 2015, leading to 429,000 deaths, most of which were children aged under five years in Africa.

Although malaria vaccines are currently not available (see, for instance, [10]), there are a number of drugs for malaria treatment. These drugs have varying degrees of efficacy [11]. As reported in [12], the combination of atovaquone-proguanil is reported to show high efficacy against *Plasmodium falciparum* with only modest side-effects. The effectiveness of a drug depends on its ability to clear parasites from the patient's blood. Artemisinin-based combination therapy (ACT) is currently identified as the treatment of choice for uncomplicated *Plasmodium falciparum* malaria in many parts of the world [13,14]. Another quality drug reported in [12] is MalaroneW$^{\circledR}$, a fixed-dose combination of A-P. This is reported to be highly effective and convenient with about three-day treatment courses for treating and prevention of multi-drug resistant *falciparum* malaria. Cases of poor quality antimalarial drugs have been reported to be in circulation in the markets of some African countries (see for example [13–16]). Unfortunately, the availability of these drugs can lead to treatment failure. According to the World Health Organization [17], treatment failure is defined as the inability to clear parasites from a patient's blood. The reasons for treatment failures have been generally attributed to suboptimal dosage, re-infections with a new parasite, or a point mutation in the pfcytb gene (see [6,9,12,18,19] and the references therein). One problem associated with treatment failure is that it may lead to antimalarial drug resistance (see [20]), which is defined as the ability of the parasite to survive the administration of a drug in doses equal to, or higher than, those usually recommended [18]. In general, drug resistance occurs through mutations that grant lessening of the sensitivity of a given drug or class of drugs. The consequence of this is that drug treatment will remove drug-susceptible parasites in infected humans, while resistant parasites endure.

Certain factors combine to give rise to the spread of drug resistance, although their relative contributions to resistance are not exactly known. Some of these factors include human behavior such as poor compliance, vector and parasite biology, malaria treatment failure, host immunity, pathogen superinfection resulting in within-host competition, presence of clinically-immune individuals, the number of people using the drugs, and poor drug quality [18,19,21–23]. While resistance is detrimental in the human population because it leads to death (see [6]), it is advantageous to the *Plasmodium* population because it confers a survival advantage in the presence of drugs.

Mathematical models of malaria transmission have been developed by several researchers to gain insight into the dynamics of the disease transmission so as to contribute towards its eradication. Some of these models can be found in [18,24–30], where the authors considered several features of malaria disease. There are a number of characteristics of malaria disease that complicate control effort. One of these is the existence of strains or races among the *Plasmodium* species responsible for human malaria. This was known as far back as 1920. Extensive work on strains in malaria can be found in [31] and the references therein. The races or strains can be distinguished based on their clinical virulence, infectivity, reaction to antimalarial remedies, and their antigenic properties [32]. For these reasons, many models have been developed that have considered two malaria strains based on drug-sensitive and drug-resistant malaria parasites. Examples of models that consider drug-sensitive and drug-resistant malaria dynamics can be found in [9,18,19,21,33,34]. Mathematical models for the transmission dynamics of drug-sensitive and -resistant strains can be useful in providing valuable information that will help in understanding the factors that influence the spread of drug resistance. This is important in designing rational intervention strategies for control of drug resistance and malaria transmissions in general. In the model of Tumwiine et al. [19], the authors considered the infected human population to consist of individuals with drug-sensitive and -resistant malaria strains. However, the vector population consists of only mosquitoes with

the drug-sensitive strain. In the model reported in [18], the authors neglected the disease-induced death rate, which is not realistic for malaria endemic regions. A second characteristic feature of malaria that complicates eradication is the occurrence of backward bifurcation, which is a situation where the locally-asymptotically-stable disease-free state co-exists with a locally-asymptotically-stable endemic equilibrium point. This phenomenon has been observed in many epidemic models. See, for instance, [9,35–39]. In this scenario, the requirement for the basic reproduction number to be less than one for the disease to be eradicated no longer holds. In many situations, models with disease-induced mortality exhibit backward bifurcation. However, there are models with disease-induced mortality where backward bifurcations have not been reported. Hadeler and Van den Driessche [40] have shown that even without the disease-induced death rate, backward bifurcation can occur. From the work of Feng et al. [41], the authors reported that the existence of backward bifurcation depends on the choice of incidence function. Baba et al. [42] studied a general two-strain epidemic model with linear incidence functions and found no backward bifurcation. Similarly, in the model of Baba and Evren reported in [43], the authors considered a linear incidence function for one strain and a non-linear incidence function in the other strain, and no backward bifurcation was found. The third characteristic of malaria disease that complicate control efforts is clinical immunity, which is a situation where protection against the clinical symptoms of the disease are developed despite the presence of the parasites [37,44,45]. The fourth feature is seasonality. In the context of malaria transmission, seasonality encapsulates complex phenomenon whose definition varies in many studies. Temperature variations have been reported by many to play a significant role in the dynamics of malaria transmissions. For example, the report of Roll Back Malaria 2015 indicated that a rise in temperature by 2–3 °C will increase the number of people at climatic risk of malaria by 3–5%. Furthermore, the abundance of mosquitoes and the transmission risk have been reported to be influenced by temperature [46–49]. At high temperatures, studies have indicated that people are unlikely to use ITNs much [50]. See the following for models related to seasonality in malaria transmission [46,48,49,51–55].

The use of ITNs is one of the common forms of protection against malaria transmission. According to the reports in [25,50,56,57], ITN possession does not translate into use, and the efficacy of ITNs could wane over time due to frequent washing and exposure to direct sunlight, among other reasons. Despite these drawbacks, ITNs usage was rated among the most important strategies against malaria transmissions [38,58,59]. One key constraint in sustainable use of ITNs is the need for regular re-treatment, otherwise they lose their efficiencies. In the models reported in [4,50], the authors introduced an explicit equation for mosquito biting rates as a linear decreasing function of the proportion of ITN usage. Although this a plausible approach, the biting rate function does not contain a parameter that will mimic the efficacy of ITNs. Moreover, Lunde et al. [48] showed that the biting rate is a non-linear function of temperature.

This study presents a deterministic model for studying the dynamics of malaria transmission in the presence of antimalarial drug resistance using an autonomous system of differential equations. We divided the infected vector population into two compartments viz. those infected with drugs-sensitive and those infected with drug-resistant strains. Furthermore, the recovered human population is divided into recovered with drug-sensitive and recovered with drug-resistant strains. The new insight that can be obtained can help policy makers in designing effective malaria control measures. The current study extends the work of Tumwiine et al. and Agusto et al. [19,50] by:

1. Proposing a simple model of the mosquito biting rate as a non-linear function of ITN usage and including a parameter in the function that represents ITN efficacy. This will form the basis for studying ITN usage and its efficacy.
2. Investigating a wide range of intervention strategies through global sensitivity analysis to determine the impacts of drug treatment and its efficacy and ITN usage and its efficacy in controlling malaria.
3. Conducting a global sensitivity analysis to determine the influence of ITN usage, drug treatment, and their efficacies and other model parameters on the dynamics of malaria transmission. This could

help in devising optimal intervention strategies that will offer more realistic predictions towards controlling malaria's spread.

The paper is organized as follows. We formulate the model in Section 2 and analyze it qualitatively in Section 3. In Section 4, we conduct a numerical simulation of the model equations. In Section 5, we conduct global uncertainty and sensitivity analysis. Section 6 is the discussion part, and in Section 7, we present our conclusions.

## 2. Model Formulation

In this section, we formulate a mathematical model for the transmission and spread of malaria using compartments in which individuals move between susceptible, infected, and immune classes in the human population and between susceptible and infected classes in the mosquito population, respectively. The model is formulated by neglecting the exposed classes for two reasons: (1) reducing the number of compartments to make the model analysis easier; (2) following Agusto et al. [50], we assume that the disease is fast advancing, so that the exposed stage is minimal and therefore neglected. The total human and mosquito populations are denoted by $N_h(t)$ and $N_v(t)$, respectively. As reported in [50], malaria transmission is a decreasing function of bed-net usage, and consequently, the authors assumed a linear decreasing function to represent the mosquito-human biting rate given by $\beta(b) = \beta_{max} - b\left(\beta_{max} - \beta_{min}\right)$. In this report, we propose a non-linear decreasing biting rate function of the proportion of ITN usage $b$ and its efficacy $\gamma$ defined by:

$$a = \frac{\beta_{\max}\beta_{\min}}{\left(\beta_{\max} - \beta_{\min}\right)b\gamma + \beta_{\min}}, \tag{1}$$

where $0 \le b, \gamma \le 1$, $\beta_{\max}$, $\beta_{\min}$ are the maximum and minimum biting rates, respectively. Bed-nets are generally used at certain times of the night; hence, we follow the assumption made in [50] that even if the entire host population used fully-efficient bed-nets ($b = 1, \gamma = 1$), the transmission can only be reduced to a minimum value $\beta_{min} > 0$. Likewise, if nobody uses bed-nets ($b = 0$), transmission will be at its maximum level. The motivation for the choice of (1) emanates from some reports that a drastic decline in the disease transmission was witnessed in some parts of Africa due to ITN usage; see [50] and the references therein. The authors further suggested that using a linear decreasing function is simply a simplifying assumption and that an exponentially-decreasing function or other functions that decrease faster might provide a better estimate of the biting rate. Note that Formula (1) provided us with the basis for studying the efficacy of ITN, which is absent in the work of Agusto et al. [50]. The elasticity index of $a$ with respect to $\gamma$ is given by:

$$\frac{\gamma}{a}\frac{\partial a}{\partial \gamma} = -\frac{\left(\beta_{\max} - \beta_{\min}\right)b\gamma a}{\left(\beta_{max} - \beta_{\min}\right)b\gamma a + \beta_{\min}} \le 0, \tag{2}$$

which clearly indicates that increasing ITN efficacy will decrease the biting rate. We assume that disease transmission via the biting rate is the same for the human and mosquito populations and is given by Equation (1).

*2.1. Model Structure*

2.1.1. Human Dynamics

The total human population $N_h(t)$ is divided into five compartments by modifying the model of Tumwiine et al. [18,19]. The compartments are $S_h(t)$, $I_{hs}(t)$, $I_{hr}(t)$, $R_{hs}(t)$, $R_{hr}(t)$, which represent the sizes of the susceptible, infected with the sensitive strain, infected with the resistant strain, recovered with the

sensitive strain, and recovered with the resistant strain categories, respectively. All recruitment was assumed to be into the susceptible human population generated via birth and/or immigration at a rate $\psi_h N_h$. This population is further increased by the loss of acquired immunity by individuals from the recovered classes at the rate $\alpha_s$ (sensitive strain) and $\alpha_r$ (resistant strain), respectively, and from the inflow of individuals that are treated successfully from the infected class with the sensitive strain at the rate $\epsilon T_s$. The susceptible human population is decreased by natural death at the rate $\phi_h = (\mu_{h2} N_h + \mu_h)$ and by the force of infection following effective contacts with infected mosquitoes, denoted by $\Lambda_h = \frac{ab_1 (I_{vs} + I_{vr})}{N_h}$. Here, $\mu_{h2}, \mu_h$ represents the density-dependent and density-independent parts of the human death rate and emigration, respectively, $b_1$ is the probability of infection from mosquito to human giving that there is contact.

The population of individuals in the infected class with the sensitive strain is increased by the proportion of susceptible humans $(\beta)$ who become infected. It is decreased by natural death rate $\phi_h$, disease-induced death rate $\delta_h$, individuals who recovered from this class at a rate $r_s$, and individuals who are treated at the rate $T_s$.

The population of individuals in the infected class with the resistant strain is increased by the proportion of susceptible humans $(1 - \beta)$ who become infected and the inflow of infected individuals with the sensitive strain whose treatment fails at a rate $(1 - \epsilon) T_s$. It is decreased by the natural and disease-induced death at the rate $\phi_h, \delta_h$, respectively individuals who recovered from this class at a rate $r_r$.

The population of individuals in the recovered class with the sensitive strain is increased by individuals who recover with immunity from the infected class with the sensitive strain at a rate $r_s$ and is decreased by the combination of individuals from this class who lose immunity at a rate $\alpha_s$ and natural death rate $\phi_h$.

The population of individuals in the recovered class with the resistant strain is increased by individuals who recover with immunity from the infected class with the resistant strain at a rate $r_r$ and is decreased by the combination of individuals from this class who lose immunity at a rate $\alpha_r$ and natural death rate $\phi_h$.

### 2.1.2. Mosquitoes' Dynamics

The total mosquito population denoted by $N_v(t)$ has three epidemiological classes denoted by $S_v(t)$, $I_{vs}(t)$, $I_{vr}(t)$, which represent the sizes of susceptible, infected with the sensitive strain, and infected with the resistant strain classes, respectively.

The population of the susceptible mosquito is generated by recruitment through birth at a rate $\psi_v N_v$ and is decreased by the force of infection denoted by $\Lambda_v = \frac{ab_2 (I_{hs} + I_{hr})}{N_h}$, where $b_2$ is the probability of infection from infected humans to susceptible mosquitoes. This population is reduced by natural death $\phi_v = \mu_{v2} N_v + \mu_v$ and death following contact with ITNs at the rate $b\gamma\mu_{v3}$. Here, $\mu_{v2}, \mu_v$ represents the density-dependent and density-independent parts of the mosquitoes' death rate, respectively, and $\mu_{v3}$ represents the ITN-induced death rate.

The population of infected mosquitoes with the sensitive strain is generated by the proportion $(1 - \alpha)$ of infected vectors. It is decreased by natural death rate $\phi_v$.

The population of infected mosquitoes with the resistant strain is generated by the proportion $(\alpha)$ of infected vectors. It is decreased by natural death rate $\phi_v$.

### 2.2. The Model

It follows, based on the above derivations and assumptions, that the model for the transmission dynamics of malaria is given by the following deterministic system of non-linear differential equations.

The flow diagram of the model is depicted in Figure 1, and the state variables and parameters of the model are described in Tables 1 and 2, respectively:

$$
\begin{aligned}
\frac{dS_h}{dt} &= \psi_h N_h + \epsilon\, T_s I_{hs} + \alpha_s R_{hs} + \alpha_r R_{hr} - \phi_h\, S_h - \Lambda_h S_h, \\
\frac{dI_{hs}}{dt} &= \beta \Lambda_h S_h - I_{hs}\left(\phi_h + T_s + \delta_h + r_s\right), \\
\frac{dI_{hr}}{dt} &= (1-\beta)\,\Lambda_h S_h + (1-\epsilon)\,T_s I_{hs} - I_{hr}\left(r_r + \delta_h + \phi_h\right), \\
\frac{dR_{hs}}{dt} &= r_s I_{hs} - \left(\phi_h + \alpha_s\right) R_{hs}, \\
\frac{dR_{hr}}{dt} &= r_r I_{hr} - \left(\phi_h + \alpha_r\right) R_{hr}, \\
\frac{dS_v}{dt} &= \psi_v N_v - \Lambda_v S_v - \phi_v S_v, \\
\frac{dI_{vs}}{dt} &= (1-\alpha)\,\Lambda_v S_v - \phi_v I_{vs}, \\
\frac{dI_{vr}}{dt} &= \alpha\,\Lambda_v S_v - \phi_v I_{vr},
\end{aligned}
\tag{3}
$$

together with the total human and vector populations given by:

$$
\frac{dN_h}{dt} = \psi_h N_h - \delta_h\left(I_{hs} + I_{hr}\right) - \phi_h\, N_h,
\tag{4}
$$

$$
\frac{dN_v}{dt} = \psi_v N_v - \phi_v N_v - b\gamma\mu_{v3}.
\tag{5}
$$

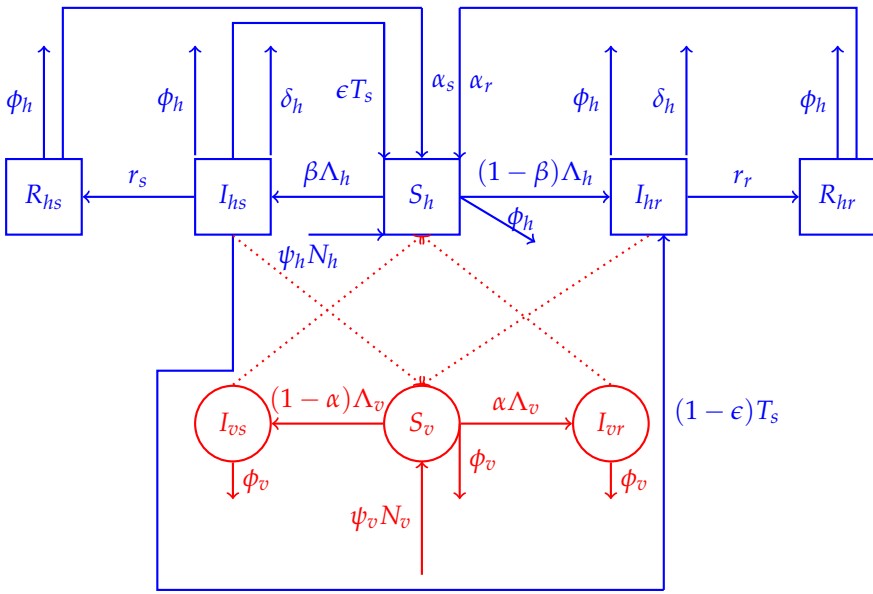

**Figure 1.** Susceptible humans $S_h$ can be infected by infectious mosquitoes. They then move to the respective infectious compartments, $I_{hs}, I_{hr}$, before entering the recovered classes $R_{hs}, R_{hr}$, respectively. Recovered humans can enter the susceptible class again or die. The susceptible mosquitoes, $S_v$, can become infected when they bite infectious humans. The infected mosquitoes then move to one of the infected classes $I_{vs}, I_{vr}$.

**Table 1.** Description of the state variable of the model.

| Variable | Description |
|---|---|
| $S_h$ | Population of susceptible humans |
| $I_{hs}$ | Population of infected humans with the sensitive strain |
| $I_{hr}$ | Population of infected humans with the resistant strain |
| $R_{hs}$ | Population of recovered humans with the sensitive strain |
| $R_{hr}$ | Population of recovered humans with the resistant strain |
| $N_h$ | Total human population |
| $S_v$ | Population of susceptible mosquitoes |
| $I_{vs}$ | Population of infected mosquitoes with the sensitive strain |
| $I_{vr}$ | Population of infected mosquitoes with the resistant strain |
| $N_v$ | Total population of mosquitoes |

**Table 2.** Parameters and their descriptions. ITN, insecticide-treated bed-net.

| Parameters | Description and Dimension |
|---|---|
| $\psi_h$ | Recruitment rate into human population (humans $\times$ day$^{-1}$) |
| $\psi_v$ | Recruitment rate into the mosquitoes' population (mosquitoes $\times$ day$^{-1}$) |
| $T_s$ | Treatment rate of infected humans with the sensitive strain (day$^{-1}$) |
| $a$ | Average daily biting rate by a single mosquito of humans (day$^{-1}$) |
| $b_2$ | Probability of transmission of infection from infected humans to susceptible mosquitoes |
| $b_1$ | Probability of transmission of infection from infected mosquitoes to susceptible humans |
| $\rho_1$ | Number of mosquitoes per human host |
| $b$ | Proportion of ITN usage |
| $\beta_{max}$ | Maximum biting rate per mosquito (day$^{-1}$) |
| $\beta_{min}$ | Minimum biting rate per mosquito (day$^{-1}$) |
| $\epsilon$ | Treatment efficacy (day$^{-1}$) |
| $\gamma$ | ITN efficacy (day$^{-1}$) |
| $\alpha_s$ | Rate at which humans with sensitive strains lose immunity (day$^{-1}$) |
| $\alpha_r$ | Rate at which humans with resistant strains lose immunity (day$^{-1}$) |
| $\alpha$ | Proportion of infected vectors that developed resistance |
| $r_r$ | Rate at which humans with resistant strains acquire immunity (day$^{-1}$) |
| $r_s$ | Rate at which humans with sensitive strains acquire immunity (day$^{-1}$) |
| $\beta$ | Proportion of susceptible humans who become infected with the sensitive strain |
| $\delta_h$ | Disease-induced death rate for infected humans (day$^{-1}$) |
| $\mu_{h2}$ | Density-dependent part of the death and emigration rate for humans (human $\times$ day$^{-1}$) |
| $\mu_h$ | Density-independent part of the death rate for humans (human $\times$ day$^{-1}$) |
| $\mu_v$ | Density-independent part of the death rate for mosquitoes (mosquitoes $\times$ day$^{-1}$) |
| $\mu_{v2}$ | Density-dependent part of the death rate for mosquitoes (day$^{-1}$) |
| $\mu_{v3}$ | ITN-induced death rate for mosquitoes (day$^{-1}$) |

The state variables of the model are given in Table 1, while Table 2 gives the model parameters and their descriptions. Our model provides a basis for studying treatment and ITN efficacy.

## 3. Basic Properties of the Model

In this section, the mathematical analysis of Model (3), will be explored.

### 3.1. Basic Properties of the Model

**Lemma 1.** *Let $k_h, k_v$ be the carrying capacities of human and mosquito populations, respectively. The closed set*
$D = \{(S_h, I_{hs}, I_{hr}, R_{hs}, R_{hr}, S_v, I_{vr}, I_{vs}) \in R_+^8 : N_h \leq k_h, N_v \leq k_v\}$

is positively invariant and attracting.

**Proof.** Adding the first five equations and the last three equations of Model (3), we obtain:

$$\frac{dN_h}{dt} = \psi_h N_h - \delta_h (I_{hs} + I_{hr}) - \phi_h N_h,$$

$$\frac{dN_v}{dt} = \psi_v N_v - \phi_v(N_v) N_v.$$

The mosquito and human populations are modeled by logistic growth with carrying capacity $k_v = \frac{\psi_v - \mu_v - \mu_{v3} b\gamma}{\mu_{v2}}$, $k_h = \frac{\psi_h - \mu_h}{\mu_{h2}}$, respectively. It is easy to see that $\frac{dN_h}{dt} \leq (\psi_h - \mu_h) N_h \left(1 - \frac{N_h}{k_h}\right)$ and $\frac{dN_v}{dt} \leq (\psi_v - \mu_v - b\gamma\mu_{v3}) N_v \left(1 - \frac{N_v}{k_v}\right)$. This shows that $\frac{dN_h}{dt} \leq 0$ if $\frac{N_h}{k_h} \geq 1$, and it approaches $k_h$. Similarly, $\frac{dN_v}{dt} \leq 0$ if $\frac{N_v}{k_v} \geq 1$, and it approaches $k_v$. Hence, using comparison theory (see [9] and the references therein),

$$N_h(t) \leq \frac{k_h N_h(0)}{N_h(0)\left(1 - e^{-(\psi_h - \mu_h)t}\right) + k_h e^{-(\psi_h - \mu_h)t}},$$

$$N_v(t) \leq \frac{k_v N_v(0)}{N_v(0)\left(1 - e^{-(\psi_v - \mu_v - \mu_{v3}b\gamma)t}\right) + k_v e^{-(\psi_v - \mu_v - \mu_{v3}b\gamma)t}}.$$

If $N_h(0) \leq k_h$, then $N_h(t) \leq k_h$, and if $N_v(0) \leq k_v$, then $N_v(t) \leq k_v$. Thus, the region $D$ is positively invariant for the model. Moreover, if $N_h(0) \geq k_h, N_v(0) \geq k_v$, then either the solution enters the region $D$ in finite time or $N_h(t) \to k_h, N_v(t) \to k_v$, as $t \to \infty$. Thus, the region attracts all solutions in $R_+^8$. Now that we have shown that $D$ is positively invariant, the requirement for the existence and uniqueness of solutions holds for Model (3) (see [9]). □

*3.2. Possibility of Backward Bifurcation*

In this section, we will investigate the possibility of backward bifurcation in Model (3). The equilibrium solutions of the total human and mosquito population are important in determining the endemic equilibrium point of the model. The mosquitoes' population has a positive equilibrium solution $N_v^*$ if $\psi_v - \mu_v - b\gamma\mu_{v3} > 0$.

**Theorem 1.** *If $N_v^*$ exist, then the number of endemic equilibrium points of the malaria model (3) is equivalent to the number of positive roots of one of the polynomials in $N_h^*$:*

$$
\begin{aligned}
P_{RM} =\ & B_2 \mu_{h2}{}^2 N_h{}^{*3} - B_2 \mu_{h2} (B_2 F_3 \delta_h - 2\mu_h + \psi_v) N_h{}^{*2} \\
& - (B_2 \mu_h (B_2 F_3 \delta_h - \mu_h + \psi_v) + K_1 \mu_{h2}) N_h^* + K_1 \psi_v - F_3 K_2 \delta_h - K_1 \mu_h,
\end{aligned}
\tag{6}
$$

$$
\begin{aligned}
P_{RN} =\ & B_3 \mu_{h2}{}^2 N_h{}^{*3} - B_3 \mu_{h2} (B_2 F_3 \delta_h - 2\mu_h + \psi_v) N_h{}^{*2} \\
& + (F_2 \mu_{h2} - B_3 \mu_h (B_2 F_3 \delta_h - \mu_h + \psi_v)) N_h^* + F_1 F_3 \delta_h + F_2 \mu_h - F_2 \psi_v,
\end{aligned}
\tag{7}
$$

*provided $R_N > 1$ or $R_M > 1$,*

where

$$R_M = \frac{\beta A_1 B_4 N_v^* \alpha_1 \phi_v^* \psi_v}{\phi_h^* B_2 N_h^*}, \quad R_N = \frac{A_1 B_1 B_5 N_v^* \alpha_1 \beta_1 \psi_v^*}{\phi_h^* B_2 B_3 N_h^*},$$

$$B_4 = \delta_h{}^2 + ((1-\beta) r_s + (-\beta\epsilon + 1) T_s + \beta r_r + 2\phi_h) \delta_h + (1-\beta)\phi_h^* r_s + (-\beta\epsilon + 1)\phi_h^* T_s + \beta r_r \phi_h^* + \phi_h^{*2},$$

$$B_5 = A_1 \beta_1 (\beta T_2 \psi_h + T_1 \beta_1 \psi_h), \quad R_R = \phi_h^* + \delta_h + r_r, \quad T_1 = \phi_h^* + T_s + \delta_h + r_s, \quad T_2 = (1-\epsilon) T_s,$$

$$B_1 = A_2 (\beta R_R + \beta T_2 + T_1 \beta_1), \quad B_2 = \beta T_2 \phi_v^* + T_1 \beta_1 \phi_v^*, \quad B_3 = (R_R + T_2)(\beta T_2 + T_1 \beta_1),$$

$$S_1 = \frac{r_s}{\alpha_s}, \quad S_2 = \frac{r_r}{\alpha_r}, \quad A_1 = a_1 b_1, \quad A_2 = a_2 b_2,$$

$$F_1 = A_1 B_1 B_5 N_v \alpha_1 \beta_1 \psi_v, \quad F_2 = A_1 B_4 N_v \alpha_1 \beta_1 \psi_v, \quad F_3 = \frac{\beta R_R + \beta T_2 + T_1 \beta_1}{B_1 (\beta T_2 + T_1 \beta_1)},$$

$$K_1 = \beta A_1 B_4 N_v \alpha_1 \phi_v^* \psi_v, \quad K_2 = \beta A_1 B_1 B_5 N_v \alpha_1 \phi_v^* \psi_v.$$

To find the equilibrium point of (3), we set the right-hand side of the equations to zero. We then express $S_h^*$, in terms of $I_{vs}^*$, $S_v^*$, and $I_{hr}^*$, $S_v^*$ in terms of $I_{hr}^*$, $I_{hs}^*$ in terms of $I_{hr}^*$, $R_{hs}^*$ in terms of $I_{hs}^*$, and $R_{hr}^*$ in terms of $I_{vr}^*$ as follows:

$$I_{vr}^* = \frac{\alpha S_v^* A_2 \left(I_{hs}^* + I_{hr}^*\right)}{N_h^* \phi_v^*},$$

$$I_{hs}^* = \frac{I_{hr}^* R_R \beta}{\beta T_2 + T_1 \beta_1},$$

$$S_h^* = \frac{I_{hr}^* N_h^{*2} R_R T_1 \phi_v^*}{\alpha B_1 I_{hr}^* S_v^* + B_2 N_h^* I_{vs}^*},$$

$$S_v^* = \frac{N_h^* N_v^* \psi_v (\beta T_2 + T_1 \beta_1)}{B_1 I_{hr}^* + B_2 N_h^*}, \tag{8}$$

$$R_{hs}^* = S_1 I_{hs}^*, \tag{9}$$

$$R_{hr}^* = S_2 I_{hr}^*. \tag{10}$$

Similarly, we express $I_{vs}^*$ as:

$$I_{vs}^* = \begin{cases} \frac{I_{hr}^* \phi_h^* N_h^* B_2}{A_1 \beta \left(B_4 I_{hr}^* - B_5 N_h^*\right) \phi_v^*}, & if\, A_1 \beta \left(B_4 I_{hr}^* - B_5 N_h^*\right) \phi_v^* > 0 \\ \frac{I_{hr}^* \phi_h^* N_h^* B_3}{A_1 \beta_1 \left(-B_4 I_{hr}^* + B_5 N_h^*\right)}, & if\, A_1 \beta \left(B_4 I_{hr}^* - B_5 N_h^*\right) \phi_v^* < 0. \end{cases} \tag{11}$$

Depending on which condition in (11) holds, we then solved for $I_{hr}^*$ explicitly in terms of parameters as:

$$I_{hr}^* = \begin{cases} \frac{N_h^{*2}(R_N-1)\phi_h^* B_2 B_3}{B_1 \left(A_1 B_4 N_v^* \alpha_1 \beta_1 \psi_v + \phi_h^* B_3 N_h^*\right)}, & \text{if the first condition in (11) holds} \\ \frac{\beta A_1 B_1 B_5 N_v^* \alpha_1 \phi_v^* \psi_v + \phi_h^* B_2^2 N_h^*}{\phi_h^* B_2 B_1 (R_M-1)}, & \text{if the second condition in (11) holds,} \end{cases} \tag{12}$$

Using Equation (12), we find an explicit expression of $I_{hs}^*$ from the second equation in (9) in terms of the parameters alone. We then substitute for $I_{hs}^*$ and $I_{hr}^*$ in Equation (4) to obtain cubic polynomials in $N_h^*$ as:

$$\begin{aligned} P_{RM} =\ & B_2 \mu_{h2}^2 N_h^{*3} - B_2 \mu_{h2} \left(B_2 F_3 \delta_h - 2\mu_h + \psi_v\right) N_h^{*2} \\ & - \left(B_2 \mu_h \left(B_2 F_3 \delta_h - \mu_h + \psi_v\right) + K_1 \mu_{h2}\right) N_h^* + K_1 \psi_v - F_3 K_2 \delta_h - K_1 \mu_h, \end{aligned} \tag{13}$$

$$\begin{aligned} P_{RN} =\ & B_3 \mu_{h2}^2 N_h^{*3} - B_3 \mu_{h2} \left(B_2 F_3 \delta_h - 2\mu_h + \psi_v\right) N_h^{*2} \\ & + \left(F_2 \mu_{h2} - B_3 \mu_h \left(B_2 F_3 \delta_h - \mu_h + \psi_v\right)\right) N_h^* + F_1 F_3 \delta_h + F_2 \mu_h - F_2 \psi_v. \end{aligned} \tag{14}$$

It is possible to have zero, one, two, or three positive values of $N_h^*$, and for each of these values, we can find the corresponding equilibrium point for Model (3) provided the conditions of Theorem 1 are satisfied. Theorem 1 provides the possibilities for backward bifurcation because the existence of multiple endemic equilibria may lead to one of them co-existing with the disease-free equilibrium point when the reproduction number of the full model is less than unity. We have not explored this possibility any further in this work.

### 3.3. Scaling and Non-Existence of Backward Bifurcation

To analyze the malaria model (3), we think it is easier to work with a fractional population instead of actual populations by scaling the population of each class by the total species population. We let: $S_v = zN_v, I_{vr} = hN_v, S_h = uN_h, I_{vs} = gN_v, R_{hs} = xN_h, R_{hr} = yN_h, I_{hr} = wN_h, I_{hs} = vN_h$. Following [27], we scaled the total human and vector populations in Model (3) using their respective carrying capacities as $N_h = k_h N_h^*, N_v = k_v N_v^*$. After simplifying, we have a six-dimensional system of equations as:

$$
\begin{aligned}
\frac{dv}{dt} &= \beta \xi_1 (1 - w - v - x - y)(g + h) - vp_1, \\
\frac{dw}{dt} &= \xi_1 (1 - w - v - x - y)((1 - \beta)(g + h)) - wp_2 + vp_3, \\
\frac{dx}{dt} &= r_s v - xc_1, \\
\frac{dy}{dt} &= r_r w - yc_2, \\
\frac{dg}{dt} &= k_1 (1 - g - h)((1 - \alpha)(v + w)) - g\psi_v, \\
\frac{dh}{dt} &= k_1 (1 - g - h)\alpha (v + w) - h\psi_v,
\end{aligned}
\tag{15}
$$

where $p_1 = (r_s + \delta_h + T_s + \psi_h)$, $p_2 = (r_r + \delta_h + \psi_h)$, $p_3 = T_s (1 - \epsilon)$, $c_1 = (\alpha_s + \psi_h)$, $c_2 = (\alpha_r + \psi_h)$, and $\xi_1(N_h, N_v) = \frac{ab_1 N_v}{N_h} = \frac{ab_1 k_v \rho_1}{k_h}$, where $k_1 = ab_2, \rho_1 = \frac{N_v^*}{N_h^*}$, in line with the work of Ngwa and Shu [27]. It is sufficient to study Model (15) in the subspace $\Omega \times [0, \infty)$ of $R_+^6$ where $\Omega = \{v, w, x, y, g, h : 0 \leq v, w, x, y, g, h < 1, 0 \leq v + w + x + y < 1, g + h < 1\}$. The dynamics of the system in the region $\Omega$ will henceforth be investigated.

### 3.4. Stability of the Disease-Free Equilibrium Point

The model (15) has a disease-free equilibrium (DFE) given by $E_{DFE} = (v^*, w^*, x^*, y^*, g^*, h^*) = (0, 0, 0, 0, 0, 0)$. By using the next-generation operator method, the non-negative matrix, $F$, of the infection terms and the non-singular matrix $V$, of the transition terms, are respectively obtained as:

$$
F = \begin{bmatrix}
0 & 0 & 0 & 0 & \beta \xi_1 & \beta \xi_1 \\
0 & 0 & 0 & 0 & \xi_1 (1 - \beta) & \xi_1 (1 - \beta) \\
0 & 0 & 0 & 0 & 0 & 0 \\
0 & 0 & 0 & 0 & 0 & 0 \\
k_1 (1 - \alpha) & k_1 (1 - \alpha) & 0 & 0 & 0 & 0 \\
\alpha k_1 & \alpha k_1 & 0 & 0 & 0 & 0
\end{bmatrix},
\tag{16}
$$

$$
V = \begin{bmatrix}
p_1 & 0 & 0 & 0 & 0 & 0 \\
-p_3 & p_2 & 0 & 0 & 0 & 0 \\
-r_s & 0 & c_1 & 0 & 0 & 0 \\
0 & -r_r & 0 & c_2 & 0 & 0 \\
0 & 0 & 0 & 0 & \psi_v & 0 \\
0 & 0 & 0 & 0 & 0 & \psi_v
\end{bmatrix}
\tag{17}
$$

$$
\rho = \pm \frac{\sqrt{-p_2 p_1 \psi_v \xi_1 k_1 \left( \beta \, p_1 + \beta - p_3 - \beta \, p_2 - p_1 \right)}}{p_2 p_1 \psi_v},
\tag{18}
$$

where $\rho$ is the spectral radius of the next-generation matrix. We define the effective reproduction number $R_{eff}$ as:

$$
R_{eff} = \frac{\xi_1 k_1 \left( 1 - \beta \right)}{p_2 \psi_v} + \frac{k_1 \xi_1 \beta \left( p_2 + p_3 \right)}{p_2 p_1 \psi_v}.
\tag{19}
$$

The formula in Equation (19) is made up of two parts: the reproduction number due to drug-resistant and the reproduction number due to drug-sensitive strains, which might be written as:

$$
R_{eff} = R_r + R_s,
\tag{20}
$$

where $R_r = \frac{\xi_1 k_1 (1-\beta)}{(r_r + \delta_h + \psi_h) \psi_v}$, $R_s = \frac{\xi_1 k_1 \beta \, (r_r + \delta_h + \psi_h + T_s (1-\epsilon))}{(r_r + \delta_h + \psi_h)(r_s + \delta_h + T_s + \psi_h) \psi_v}$. Hence, using Theorem 2 of [60], we have established the following result:

**Lemma 2.** *The disease-free equilibrium $E_{DFE}$ of Model* (15) *is locally-asymptotically stable (LAS) if $R_{eff} < 1$, and unstable if $R_{eff} > 1$.*

The threshold quantity $R_{eff}$ measures the average number of secondary cases generated by a single infected individual in a population where some infected individuals are treated and some are using ITNs. In the absence of treatment and ITN usage, we set $T_s = 0$, $b = 0$, and $\xi_1 \equiv \xi_2$ with $a = \beta_{max}$ in the formula for $\xi_1$. Hence, $\xi_2 > \xi_1$. The effective reproduction number reduces to the basic reproduction number denoted by $R_{eff}^{00}$ where:

$$
R_{eff}^{00} = \frac{\xi_2 k_1 \left( 1 - \beta \right)}{(r_r + \delta_h + \psi_h) \, \psi_v} + \frac{\xi_2 k_1 \beta}{(r_s + \delta_h + \psi_h) \, \psi_v}.
\tag{21}
$$

Lemma 2 shows that malaria can be eradicated if the initial populations of the model are confined in the basin of attraction of the DFE. To guarantee the elimination of disease irrespective of initial population sizes, a global stability analysis of the DFE is required. This is done below, using the Lyapunov function.

*3.5. Global Stability of the DFE*

**Theorem 2.** *Assume $R_{eff} < 1$, then the disease-free equilibrium point $E_{DEF}$ of Model* (15) *is globally asymptotically stable.*

**Proof.** Consider the Lyapunov function:

$$
F = L_1 v + L_2 w + L_3 (x + y) + L_5 (g + h),
\tag{22}
$$

where $L_5 = \xi_1 (p_1 r_r (1 - \beta) + \beta p_2 r_s + \beta p_3 r_r)$, $L_3 = p_1 p_2 (1 - R_{eff})$, $L_2 = \frac{p_1 p_2 (1 - R_{eff}) r_r + L_5 k_1}{p_2}$, $L_1 = \frac{p_1 p_2 (1 - R_{eff})(p_2 r_s + p_3 r_r) + L_5 k_1 (p_2 + p_3)}{p_1 p_2}$. Some calculations shows that the time derivative becomes:

$$
\begin{aligned}
\frac{dF}{dt} = & \; -\xi_1 (L_1 \beta + L_2 \beta_1)(g + h)(x + y) - L_3 c_1 x - L_3 c_2 y \\
& + (-((L_1 \beta + L_2 \beta_1)\xi_1 + L_5 k_1)(g + h) + L_5 k_1 - L_1 p_1 + L_2 p_3 + L_3 r_s) v \\
& + (-(L_1 \beta \xi_1 + L_2 \beta_1 \xi_1 + L_5 k_1)(g + h) + L_5 k_1 - L_2 p_2 + L_3 r_r) w \\
& + \xi_1 (L_1 \beta + L_2 \beta_1)(g + h) - \psi_v L_5 (g + h).
\end{aligned}
\tag{23}
$$

We simplify (23) to get:

$$
\begin{aligned}
\frac{dF}{dt} = & \; -\xi_1 \left( \frac{(p_1 p_2 (1 - R_{eff})(p_2 r_s + p_3 r_r) + L_5 k_1 (p_2 + p_3))\beta}{p_2 p_1} \right. \\
& \left. + \frac{(p_1 p_2 (1 - R_{eff}) r_r + L_5 k_1)\beta_1}{p_2} \right)(g + h)(x + y + w + v) \\
& - p_1 p_2 \left( 1 - R_{eff} \right)(x c_1 + y c_2) - L_5 k_1 (g + h)(v + w).
\end{aligned}
\tag{24}
$$

Hence, we have $\frac{dF}{dt} < 0$ for all $(x, y, v, w, g, h) \neq (0, 0, 0, 0, 0, 0)$. Thus, by Lyapunov's theorem, the disease-free equilibrium is globally asymptotically stable. □

### 3.6. Boundary Equilibria

In this section, we investigate the existence and stability of positive equilibria involving only one strain.

### 3.6.1. Boundary Equilibria for the Drug-Sensitive Strain Only

This is an equilibrium where only the sensitive strain is present. The following results are obtained:

**Lemma 3.** *Model* (15) *has a sensitive strain type-only boundary equilibrium, given by $E_s$, whenever $R_S = \frac{k_1 \xi_1}{p_1 \psi_v} > 1$.*

**Proof.** To get an equilibrium point $E_s = (v_s^*, w_s^*, x_s^*, y_s^*, g_s^*, h_s^*) = (v_s^*, 0, x_s^*, 0, g_s^*, 0)$, of Model (15), we must have $\beta = 1, \alpha = 0, \epsilon = 1$. Substituting these values, we obtain the equilibrium point as:

$$
E_s = (v_s^*, w_s^*, x_s^*, y_s^*, g_s^*, h_s^*) = \left( \frac{(R_S - 1)\psi_h}{d_1}, 0, \frac{(R_S - 1) r_s}{d_1}, 0, \frac{\psi_h (R_S - 1)}{d_2}, 0 \right),
$$

where $d_1 = (p_1 \psi_h + \xi_1 (\psi_h + r_s)) k_1 p_1 \psi_v$, $d_2 = \xi_1 (k_1 \psi_h + \psi_v (\psi_h + r_s)) p_1 \psi_v$. This establishes that the boundary equilibrium point for the sensitive strain only exists when $R_S > 1$. □

### 3.6.2. Local Stability of $E_s$

**Lemma 4.** *$E_s$ is locally asymptotically stable.*

**Proof.** The local stability of $E_s$ is investigated by evaluating the Jacobian of Model (15) with $\beta = 1, \alpha = 0, \epsilon = 1$, and the coordinates taken in the following order $v, x, g, w, y, h$ to obtain:

$$
J(E_s) = \begin{bmatrix} M_{1s} & M_{2s} \\ \mathbf{0} & M_{4s} \end{bmatrix},
$$

where $M_{1s} = \begin{bmatrix} -\xi_1 g_s^* - p_1 & -\xi_1 g_s^* & \xi_1 (1 - v_s^* - x_s^*) \\ r_s & -c_1 & 0 \\ k_1 (1 - g_s^*) & 0 & -k_1 v_s^* - \psi_v \end{bmatrix}, M_{4s} = \begin{bmatrix} -p_2 & 0 & 0 \\ r_r & -c_2 & 0 \\ 0 & 0 & -\psi_v \end{bmatrix}$. The

eigenvalues of $J(E_s)$ can be obtained by finding the eigenvalues of the matrices $M_{1s}$ and $M_{4s}$ only. The eigenvalues of $M_{4s}$ are $-p_2, -c_2, -\psi_v$ which are all negatives. The characteristic polynomial of $M_{1s}$ is given by:

$$A = \eta^3 + B_2 \eta^2 + B_1 \eta + B_0, \tag{25}$$

where $B_2 = g_s^* \xi_1 + v_s^* k_1 + p_1 + c_1 + \psi_v$, $B_1 = (k_1 \xi_1 v_s^* + \xi_1 (c_1 + \psi_v + r_s)) g_s^* + k_1 (p_1 + c_1) v_s^* + c_1 (p_1 + \psi_v)$, $B_0 = \xi_1 (c_1 + r_s) (v_s^* k_1 + \psi_v) g_s^* + v_s^* k_1 p_1 c_1$. By the Routh–Hurwitz criterion, all the roots of Equation (25) have negative real parts if the following holds: $B_0, B_1, B_2 > 0$, and $B_1 B_2 - B_0 > 0$. Clearly, $B_i > 0, i = 0, 1, 2$. We simplify $B_1 B_2 - B_0$ to get $P_s = A_1 (g_s^*)^2 + A_2 g_s^* + A3(v_s^*)^2 + A_4 v_s^* + A_5 > 0$, where $A_1 = k_1 \xi_1^2 v_s^* + \xi_1^2 (c_1 + \psi_v + r_s)$, $A_2 = k_1^2 \xi_1 v_s^{*2} + 2 k_1 \xi_1 (p_1 + c_1 + \psi_v) v_s^* + \xi_1 (p_1 (2 c_1 + \psi_v + r_s) + c_1 (c_1 + 2 \psi_v + r_s) + \psi_v^2)$, $A_3 = k_1^2 (p_1 + c_1)$, $A_4 = k_1 (p_1 (p_1 + 2 c_1 + \psi_v) + c_1 (c_1 + 2 \psi_v))$, $A_5 = c_1 (p_1 + \psi_v) (p_1 + c_1 + \psi_v)$. It follows from the Routh–Hurwitz criterion that all the roots of the characteristic polynomial of $M_{1s}$ have negative real parts. Hence, all the roots of $J(E_s)$ have negative real parts, which establishes the stability of $E_s$. □

### 3.6.3. Boundary Equilibria for the Drug-Resistant Strain Only

This is an equilibrium where only the resistant strain is present. We state the following:

**Lemma 5.** *The model* (15) *has a resistant strain type-only boundary equilibrium, given by $E_r$, whenever $R_R = \frac{k_1 \xi_1}{p_2 \psi_v} > 1$.*

**Proof.** To get an equilibrium point $E_r = (v_r^*, w_r^*, x_r^*, y_r^*, g_r^*, h_r^*) = (0, w_r^*, 0, y_r^*, 0, h_r^*)$, of Model (15), we must have $\beta = 0, \alpha = 1$. Substituting these values, we obtain the equilibrium point as:

$$E_r = (v_r^*, w_r^*, x_r^*, y_r^*, g_r^*, h_r^*) = \left( 0, \frac{(R_R - 1) c_2}{q_1}, 0, \frac{(R_R - 1) r_r}{q_1}, 0, \frac{(R_R - 1) c_2}{q_2} \right),$$

where $R_R = \frac{k_1 \xi_1}{p_2 \psi_v}$, $q_1 = (c_2 (p_2 + \xi_1) + r_r \xi_1) k_1 p_2 \psi_v$, $q_2 = \xi_1 (c_2 (k_1 + \psi_v) + \psi_v r_r) p_2 \psi_v$. □

### 3.6.4. Local Stability of $E_r$

**Lemma 6.** *$E_r$ is locally asymptotically stable.*

**Proof.** The local stability of the $E_r$ is investigated by evaluating the Jacobian of Model (15) with $\beta = 0$, $\alpha = 1$, and the coordinates taking in the following order $v, x, g, w, y, h$ to obtain:

$$J(E_r) = \begin{bmatrix} M_{1r} & \mathbf{0} \\ M_{3r} & M_{4r} \end{bmatrix},$$

$$\text{where} \quad M_{1r} = \begin{bmatrix} -p_1 & 0 & 0 \\ r_s & -c_1 & 0 \\ 0 & 0 & -\psi_v \end{bmatrix}, M_{4r} = \begin{bmatrix} -\xi_1 h_r^* - p_2 & -\xi_1 h_r^* & \xi_1\left(1 - w_r^* - y_r^*\right) \\ r_r & -c_2 & 0 \\ k_1\left(1 - h_r^*\right) & 0 & -k_1 w_r^* - \psi_v \end{bmatrix}. \quad \text{The}$$

eigenvalues of $J(E_r)$ are the eigenvalues of the matrices $M_{1r}$ and $M_{4r}$. The eigenvalues of $M_{1r}$ are $-p_1, -c_1, -\psi_v$, which are negatives. The characteristic polynomial of $M_{4r}$ is given by:

$$T = \eta^3 + D_2\eta^2 + D_1\eta + D_0, \tag{26}$$

where $D_2 = h_r^*\xi_1 + w_r^* k_1 + c_2 + p_2 + \psi_v$, $D_1 = \left(k_1\xi_1 w_r^* + \xi_1\left(c_2 + \psi_v + r_r\right)\right)h_r^* + k_1\left(c_2 + p_2\right)w_r^* + c_2\left(p_2 + \psi_v\right)$, $D_0 = \xi_1\left(c_2 + r_r\right)\left(w_r^* k_1 + \psi_v\right)h_r^* + w_r^* c_2 k_1 p_2$. Clearly, $D_i > 0, i = 0,1,2$. We simplify $D_1 D_2 - D_0$ to get $P_r = C_1(h_r^*)^2 + C_2 h_r^* + C_3(w_r^*)^2 + C_4 w_r^* + C_5$, where $C_1 = k_1\xi_1^2 w_r^* + \xi_1^2\left(c_2 + \psi_v + r_r\right)$, $C_5 = c_2\left(p_2 + \psi_v\right)\left(c_2 + p_2 + \psi_v\right)$, $C_2 = Bk_1^2\xi_1 w_r^{*2} + 2k_1\xi_1\left(c_2 + p_2 + \psi_v\right)w_r^* + \xi_1\left(c_2\left(c_2 + 2p_2 + 2\psi_v + r_r\right) + p_2\psi_v + p_2 r_r + \psi_v^2\right)$, $C_3 = k_1^2\left(c_2 + p_2\right)$, $C_4 = k_1\left(c_2\left(c_2 + 2p_2 + 2\psi_v\right) + p_2\left(p_2 + \psi_v\right)\right)$. It follows from the Routh–Hurwitz criterion that all the roots of the characteristic polynomial of $M_{4r}$ have negative real parts. Hence, all the roots of $J(E_r)$ have negative real parts, which establishes the stability of $E_r$. $\square$

### 3.7. Coexistence Equilibrium Point

Let $E_{rs} = (v^{**}, w^{**}, x^{**}, y^{**}, g^{**}, h^{**})$ be the coexistence equilibrium point of the two strains, then $E_{rs}$ is the solution of the non-linear system of equations:

$$
\begin{aligned}
0 &= r_s v^{**} - x^{**} c_1, \\
0 &= r_r w^{**} - y^{**} c_2, \\
0 &= \beta\xi_1\left(1 - w^{**} - v^{**} - x^{**} - y^{**}\right)\left(g^{**} + h^{**}\right) - v^{**} p_1, \\
0 &= \xi_1\left(1 - w^{**} - v^{**} - x^{**} - y^{**}\right)\left((1 - \beta)(g^{**} + h^{**})\right) - w^{**} p_2 + v^{**} p_3, \\
0 &= k_1\left(1 - g^{**} - h^{**}\right)(1 - \alpha)\left(v^{**} + w^{**}\right) - g^{**}\psi_v, \\
0 &= k_1\left(1 - g^{**} - h^{**}\right)\alpha\left(v^{**} + w_{**}\right) - h^{**}\psi_v.
\end{aligned}
\tag{27}
$$

We solved Equation (27) to get the components of $E_{rs}$ as:

$$
\begin{aligned}
h^{**} &= \frac{\alpha c_2 c_1 p_1 p_2 \psi_v\left(R_{eff} - 1\right)}{\Psi}, \\[2mm]
g^{**} &= \frac{c_2 c_1 p_1 p_2 \psi_v\left(R_{eff} - 1\right)(1 - \alpha)}{\Psi}, \\[2mm]
y^{**} &= \frac{\left(c_1 p_1\left(1 - \beta\right) + \beta p_3\right)c_1 p_1 p_2 \psi_v\left(R_{eff} - 1\right)}{\Phi}, \\[2mm]
x^{**} &= \frac{\beta c_1 p_1 p_2^2 \psi_v\left(R_{eff} - 1\right)c_2}{\Phi}, \\[2mm]
w^{**} &= \frac{c_2\left(-c_1 p_1\left(\beta - 1\right) + \beta p_3\right)c_1 p_1 p_2 \psi_v\left(R_{eff} - 1\right)}{\Phi}, \\[2mm]
v^{**} &= \frac{c_1^2 \beta p_1 p_2^2 \psi_v\left(R_{eff} - 1\right)c_2}{\Phi},
\end{aligned}
\tag{28}
$$

$$\tag{29}$$

where:

$$
\begin{aligned}
\Psi &= \xi_1 \left( \left( \left( \left( p_3 + p_2 \left( c_1 + 1 \right) \right) c_2 + p_3 \right) \psi_m + k_1 \left( c_1 p_2 + p_3 \right) c_2 \right) \beta + c_1 p_1 \left( 1 - \beta \right) \right) \\
&\times \left( c_2 \left( k_1 + \psi_v \right) + \psi_v \right) \right), \\
\Phi &= \left( \left( \left( c_1 + 1 \right) c_2 p_2 + p_3 c_2 + p_3 \right) \xi_1 \beta + c_1 c_2 p_1 p_2 + \xi_1 \left( 1 - \beta \right) \left( c_2 + 1 \right) p_1 c_1 \right) \\
&\times \left( c_1 p_1 \left( 1 - \beta \right) + \beta \left( c_1 p_2 + p_3 \right) \right) k_1,
\end{aligned}
$$

from which we state the following result:

**Theorem 3.** *Model* (15) *has a unique co-existence equilibrium point $E_{rs}$ whenever $R_{eff} > 1$.*

Even though we show that there is the possibility of backward bifurcation in Model (3), this phenomenon is absent in the reduced model (15). The global stability of the reduced model will now be investigated.

### 3.8. Global Stability of the Coexistence Equilibrium Point

To investigate the global stability of the equilibrium point where both the sensitive and the resistant strains coexist, we consider a special case where there is no inflow of individuals from the infected class with the sensitive strain to the infected class with the resistant strain. In this case, there is no evolution of drug resistance due to treatment failure. Hence, $\epsilon = 1$ and $p_3 = 0$. It can also be seen that if $T_s = 0$, then $p_3$ is also zero. Hence, we state the following:

**Theorem 4.** *Assuming that treatment failure does not lead to drug resistance or treatment is completely absent, then the coexistence equilibrium point is globally asymptotically stable where it exists.*

**Proof.** To prove Theorem 4, we consider the following Lyapunov function:

$$
\begin{aligned}
F_e &= \frac{G^{**} u^{**}}{v^{**} p_1} \left( v - v^{**} - v^{**} \ln \left( \frac{v}{v^{**}} \right) \right) + \frac{G^{**} u^{**}}{p_2 w^{**}} \left( w - w^{**} - w^{**} \ln \left( \frac{w}{w^{**}} \right) \right) \\
&+ \frac{W^{**} z^{**} h^{**}}{g^{**}} \left( g - g^{**} - g^{**} \ln \left( \frac{g}{g^{**}} \right) \right) + W^{**} z^{**} \left( h - h^{**} - h^{**} \ln \left( \frac{h}{h^{**}} \right) \right),
\end{aligned}
\tag{30}
$$

where $W^{**} = w^{**} + v^{**}$, $G^{**} = g^{**} + h^{**}$, $z^{**} = 1 - g^{**} - h^{**}$, $u^{**} = 1 - x^{**} - y^{**} - w^{**} - v^{**}$. Note that under the hypothesis of Theorem 3, $p_3 = 0$. After substituting the derivatives, we obtain:

$$
\begin{aligned}
\frac{dF_e}{dt} &= \frac{G^{**} u^{**}}{v^{**} p_1} \left( 1 - \frac{v^{**}}{v} \right) \left( G\beta u \xi_1 - v p_1 \right) + \frac{G^{**} u^{**}}{p_2 w^{**}} \left( 1 - \frac{w^{**}}{w} \right) \left( \xi_1 u G \beta_1 - w p_2 \right) \\
&+ \frac{W^{**} z^{**} h^{**}}{g^{**}} \left( 1 - \frac{g^{**}}{g} \right) \left( k_1 z W \alpha_1 - g \psi_v \right) + W^{**} z^{**} \left( 1 - \frac{h^{**}}{h} \right) \left( k_1 z \alpha W - h \psi_v \right),
\end{aligned}
\tag{31}
$$

where $W = v + w$, $G = g + h$, $z = 1 - g - h$, $u = 1 - x - y - v - w$, $\beta_1 = 1 - \beta$, $\alpha_1 = 1 - \alpha$. At equilibrium, we substitute $\beta = \frac{v^{**} p_1}{G^{**} u^{**} \xi_1}$, $\beta_1 = \frac{w^{**} p_2}{G^{**} u^{**} \xi_1}$, $\alpha = \frac{h^{**} \psi_v}{W^{**} z^{**} k_1}$, $\alpha_1 = \frac{g^{**} \psi_v}{W^{**} z^{**} k_1}$ in Equation (31), and after some manipulations, we obtain:

$$
\begin{aligned}
\frac{dF_e}{dt} &= \left( -\frac{g}{g^{**}} + 2 - \frac{h}{h^{**}} \right) h^{**} W^{**} z^{**} \psi_v + \left( -\frac{h^{**}}{h} + 2 - \frac{g^{**}}{g} \right) h^{**} z W \psi_v, \\
&+ uG \left( 2 - \frac{w^{**}}{w} - \frac{v^{**}}{v} \right) + \left( 2 - \frac{v}{v^{**}} - \frac{w}{w^{**}} \right) G^{**} u^{**}.
\end{aligned}
\tag{32}
$$

Further simplifications yield,

$$
\begin{aligned}
\frac{dF_e}{dt} &= -\left( W^{**} z^{**} \frac{g}{g^{**}} + W^{**} z^{**} \frac{h}{h^{**}} + zW \frac{h^{**}}{h} + zW \frac{g^{**}}{g} - 2 \left( zW + W^{**} z^{**} \right) \right) h^{**} \psi_v, \\
&\quad - \left( uG \frac{w^{**}}{w} + uG \frac{v^{**}}{v} + G^{**} u^{**} \frac{v}{v^{**}} + G^{**} u^{**} \frac{w}{w^{**}} - 2 \left( uG + G^{**} u^{**} \right) \right).
\end{aligned}
\tag{33}
$$

We claim that $\left( W^{**} z^{**} \frac{g}{g^{**}} + W^{**} z^{**} \frac{h}{h^{**}} + zW \frac{h^{**}}{h} + zW \frac{g^{**}}{g} - 2 \left( zW + W^{**} z^{**} \right) \right) \geq 0$. Note that the expression can be written as $x_1 x_3 + x_1 x_4 + \frac{x_2}{x_3} + \frac{x_2}{x_4} - 2 \left( x_1 + x_2 \right)$, where $x_1 = W^{**} z^{**}, x_2 = Wz, x_3 = \frac{g}{g^{**}}, x_4 = \frac{h}{h^{**}}$. Consider $y = A - B$ where $A, B \geq 0$ and real. The minimum value of $y$ is $-B$ and is attained when $A = 0$. Let $A = x_1 x_3 + x_1 x_4 + \frac{x_2}{x_3} + \frac{x_2}{x_4}$ $B = 2 \left( x_1 + x_2 \right)$. Since $x_3, x_4$ are non-zero (at the coexistence equilibrium point, the disease classes are non-zero). Therefore, $A = 0 \implies x_1 = x_2 = 0$. This $\implies -B = 0$. Thus, the minimum value of $x_1 x_3 + x_1 x_4 + \frac{x_2}{x_3} + \frac{x_2}{x_4} - 2 \left( x_1 + x_2 \right)$ is zero. A similar argument can be applied to the expression $\left( uG \frac{w^{**}}{w} + uG \frac{v^{**}}{v} + G^{**} u^{**} \frac{v}{v^{**}} + G^{**} u^{**} \frac{w}{w^{**}} - 2 \left( uG + G^{**} u^{**} \right) \right)$. Hence:

$$
\begin{aligned}
\left( W^{**} z^{**} \frac{g}{g^{**}} + W^{**} z^{**} \frac{h}{h^{**}} + zW \frac{h^{**}}{h} + zW \frac{g^{**}}{g} - 2 \left( zW + W^{**} z^{**} \right) \right) &\geq 0, \\
\left( uG \frac{w^{**}}{w} + uG \frac{v^{**}}{v} + G^{**} u^{**} \frac{v}{v^{**}} + G^{**} u^{**} \frac{w}{w^{**}} - 2 \left( uG + G^{**} u^{**} \right) \right) &\geq 0.
\end{aligned}
\tag{34}
$$

Thus, $\frac{dF_e}{dt} \leq 0$. It is easy to see that $\frac{dF_e}{dt} = 0$ if and only if $(v, w, x, y, g, h) = (v^{**}, w^{**}, x^{**}, y^{**}, g^{**}, h^{**})$. Thus, the set $\Phi = \{ X \in R^n | \frac{dF_e}{dt}(X) = 0 \}$ is the singleton $(v^{**}, w^{**}, x^{**}, y^{**}, g^{**}, h^{**})$. Hence, $F_e$ is a Lyapunov function. This concludes the proof. $\square$

## 4. Numerical Simulations of the Model

### 4.1. Baseline Parameter Values

Apart from theoretical results obtained from the model analysis, simulating the model equations is also important. However, finding suitable data for model simulation and sensitivity analysis is still a big challenge without an appropriate answer. Following [9,18,19,26,27,35,38,50], we used parameter values from published work in the literature to simulate and to perform sensitivity analysis of our model where such information is found. In some cases, we assumed the values of some of the parameters as shown in Table 3. The use of parameter values from many sources became necessary because we had not found a single study that incorporated all our model parameters. For example, for $r_r$ and $r_s$, we used their values reported in [18] as baseline values. However, we had not found previous studies that conducted global sensitivity analysis using these parameters. We estimated the parameter ranges by setting the lower values to be 65% and upper values to be 135% of the corresponding baseline values. We used this procedure in many instances, to estimate many parameter ranges, as shown in Table 3. In some cases, we used both estimation and reported values to provide parameter ranges. For instance, the smallest value for $\psi_v$ used in the sensitivity analysis was estimated as 65% of the baseline. We obtained the upper value from the report in [19]. The range for this parameter was also reported in [61] with the upper value being smaller than the lowest value we were using. The reason for this choice is attributed to the fact we needed a relatively higher recruitment rate to ensure that $\psi_v - \mu_v - \mu_{v3} b\gamma > 0$, which is a necessary condition for sustaining the mosquito population. The mosquitoes' mortality rate $\mu_{v3} b\gamma$, due to the use of ITNs was absent in the work of Chitnis et al. reported in [61]. Hence, a relatively smaller mosquito recruitment rate can sustain the mosquito population. Tumwiine et al. [19] varied treatment efficacy in the range [0.01 0.61]. Hence, we arbitrarily picked 0.4, which is in that interval, to represent the baseline value for $\epsilon$.

Figure 2 presents the results for simulating the malaria model (15) using the initial condition $(v, w, x, y, g, h) = (0.15, 0.05, 0.01, 0.001, 0.2, 0.6)$, for 2000 days (a) with the baseline parameter values shown in Table 3 and (b) with the same baseline values except that the treatment rate $T_s$ was increased to 0.75. Also shown on Figure 2 is the proportion of infected humans with resistant (dotted curve) and sensitive malaria strains (solid curve). The results are consistent with the reports in [18,19].

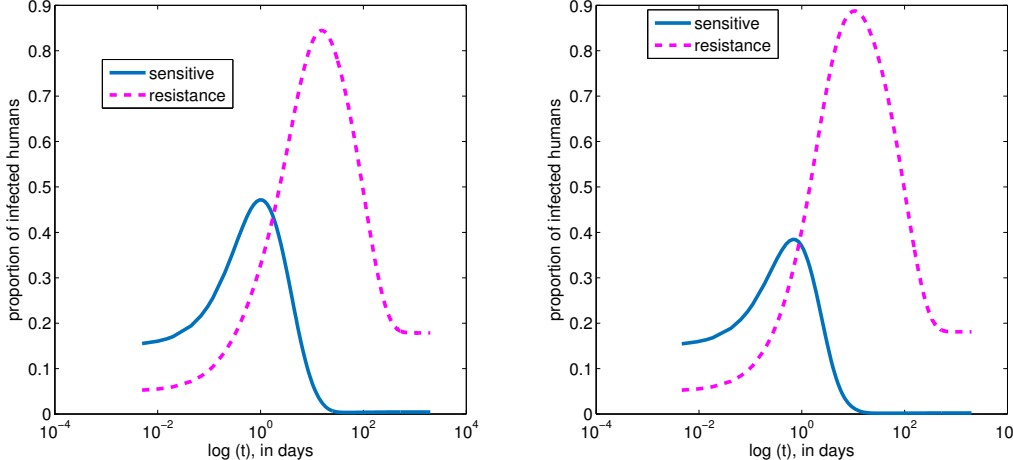

**Figure 2.** Numerical simulation of Model (15) showing the proportion of infected humans with both sensitive and resistant strains using the initial condition $(v, w, x, y, g, h) = (0.15, 0.05, 0.01, 0.001, 0.2, 0.6)$. (**a**) With the baseline values in Table 3; (**b**) With the baseline values except $T_s$ changed to 0.75.

From Figure 2, the population of the infected humans increases, reaches a maximum, and then falls. Based on the current setting, the population of the infected humans with the sensitive strain will approach zero as time progresses, and the population of humans with the resistant strain will persist. Comparing Figure 2a and 2b, the results indicate that increasing treatment without changing its efficacy tends to increase the peak value of the proportion of individuals with the resistant strain from 0.8446 to 0.8876 and decreases the peak value of the proportion of individuals with the sensitive strain from 0.4718 to 0.3846. From this finding, we can see that the relative increase in the proportion of infected humans with the resistant strain is about 5% with the corresponding decrease in the proportion of humans with the sensitive strain being about 18.5%.

## 5. Intervention Strategies and Global Sensitivity Analysis

The use of ITNs reduces the exposure of humans to mosquitoes, hence leading to reduction in the transmission of the parasites from infectious humans to susceptible mosquitoes and vice versa. Treatment reduces the number of infected individuals by reducing the parasite load of the treated humans, and hence, the transmission probability is also reduced. These strategies have the effect of reducing malaria transmission in a population. Tumwiine et al. [19] reported that in the presence of a drug-resistant population, treatment has a negative impact on the reduction of malaria transmission. In this section, we evaluate the impact of ITNs and treatment on the reproduction number by determining necessary and sufficient conditions for the effective reproduction number in our model to be smaller than the basic reproduction number.

*5.1. Analytic Intervention Strategies*

We define the reproduction numbers, in the absence of treatment and ITN ($R_{eff}^{00}$), (see Equation (21)), in the absence of treatment and the presence of ITN ($R_{eff}^{01}$), and in the presence of treatment and the absence of ITN ($R_{eff}^{10}$), respectively, as:

$$
\begin{aligned}
R_{eff}^{01} &= \frac{\xi_1 k_1 (1-\beta)}{(r_r + \delta_h + \psi_h)\, \psi_v} + \frac{\xi_1 k_1 \beta}{(r_s + \delta_h + \psi_h)\, \psi_v}, \\
R_{eff}^{10} &= \frac{\xi_2 k_1 (1-\beta)}{(r_r + \delta_h + \psi_h)\, \psi_v} + \frac{\xi_2 k_1 \beta\, (r_r + \delta_h + \psi_h + T_s(1-\epsilon))}{(r_r + \delta_h + \psi_h)\,(r_s + \delta_h + T_s + \psi_h)\, \psi_v}.
\end{aligned}
\tag{35}
$$

Using Equations (19) and (35), we get:

$$
\begin{aligned}
R_{eff}^{10} - R_{eff}^{00} &= -\frac{\xi_2 k_1 \beta\, T_s\, (\epsilon\, \delta_h + \epsilon\, \psi_h + \epsilon\, r_s + r_r - r_s)}{(r_r + \delta_h + \psi_h)\,(r_s + \delta_h + T_s + \psi_h)\, \psi_v\, (r_s + \delta_h + \psi_h)}, \\
R_{eff} - R_{eff}^{01} &= -\frac{\xi_1 k_1 \beta\, T_s\, (\epsilon\, \delta_h + \epsilon\, \psi_h + \epsilon\, r_s + r_r - r_s)}{(r_r + \delta_h + \psi_h)\,(r_s + \delta_h + T_s + \psi_h)\, \psi_v\, (r_s + \delta_h + \psi_h)}, \\
R_{eff}^{01} - R_{eff}^{00} &= -\frac{k_1\, (\beta\, r_r - \beta\, r_s + \delta_h + \psi_h + r_s)\, (\xi_2 - \xi_1)}{(r_r + \delta_h + \psi_h)\, \psi_v\, (r_s + \delta_h + \psi_h)}, \\
R_{eff} - R_{eff}^{10} &= -\frac{k_1\, (\xi_2 - \xi_1)\, (-T_s\,(\beta\,\epsilon - 1) + \beta\, r_r - r_s\,(-1+\beta) + \delta_h + \psi_h)}{(r_r + \delta_h + \psi_h)\,(r_s + \delta_h + T_s + \psi_h)\, \psi_v}.
\end{aligned}
\tag{36}
$$

Note that the last two equations of (36) are strictly negative. The remaining equations are negative whenever $\frac{r_s - r_r}{r_s + \delta_h + \psi_h} < \epsilon$. Hence, the following results are obtained:

**Theorem 5.** *If $\epsilon > \frac{r_s - r_r}{r_s + \delta_h + \psi_h}$, then $R_{eff} < R_{eff}^{01} < R_{eff}^{00}$ and $R_{eff}^{10} < R_{eff}^{00}$.*

The implication of Theorem 5 is that using any one of the two intervention strategies will have a positive impact on the reduction of the spread of malaria and that using both intervention parameters has the most positive impact. Furthermore, the basic reproduction number is guaranteed to be bigger than the reproduction numbers in the presence of one or two intervention parameters whenever the ratio of the rates at which humans with sensitive malaria strain acquire immunity to that at which humans with the resistant strain acquire immunity is less than unity. We state this as a Corollary to Theorem 5.

**Corollary 1.** *If $\frac{r_s}{r_r} \leq 1$, then $R_{eff} < R_{eff}^{01} < R_{eff}^{00}$ and $R_{eff}^{10} < R_{eff}^{00}$ are satisfied.*

The new insight that can be derived here is that in the presence of antimalarial drug resistance, treatment can have a positive impact on malaria spread. This is contrary to the report in [19].

*5.2. Numerical Intervention Strategies and Global Sensitivity Analysis*

5.2.1. Sensitivity Analysis Using Partial Rank Correlation Coefficients

In epidemic modeling, many parameters are shrouded in uncertainty, which might be due to erroneous parameter estimation and uncertainty in the exact parameter values. For these reasons, it is important to conduct sampling and sensitivity analysis to determine parameters that have a substantial influence on model output. The Sampling and Sensitivity Analysis Tools (SaSAT) is a software tool developed for such purposes (see [62]). In our model, the effective reproduction number was regulated by 20 various

malaria-related epidemiological parameters whose values varied from other studies. We assigned baseline values and ranges for each of the model parameters as discussed in Section 4.1.

**Table 3.** Parameters, their baseline values, ranges, and distribution type for sensitivity analysis.

| Parameter | Baseline Value/Source | Range/Source | Distribution for Sensitivity Analysis |
|---|---|---|---|
| $\psi_h$ | $9.3614 \times 10^{-5}$, [38] | $[6.0849, 12.17] \times 10^{-5}$, estimated | Uniform |
| $\psi_v$ | 0.4478, [38] | [0.2911, 0.7], estimated, [19] | Uniform |
| $T_s$ | 0.35, [19] | [0.2275, 0.455], estimated | Uniform |
| $b_1$ | 0.75, [19] | [0.1, 0.8], assumed | Uniform |
| $b_2$ | 0.5342, [38] | [0.072, 0.64], [56] | Uniform |
| $\rho$ | 7, assumed | [2, 8], [38] | Uniform |
| b | 0.53, [56] | [0.1325, 0.6625], estimated | Triangular, peak 0.5 |
| $\beta_{max}$ | 0.6334, [38] | [0.1, 1], [56] | Uniform |
| $\beta_{min}$ | 0.0696, [38] | [0, 0.1], [56] | Uniform |
| $\epsilon$ | 0.4, assumed | [0.01, 0.61], [19] | Uniform |
| $\gamma$ | 0.5, assumed | [0.2, 1], [58] | Uniform |
| $\alpha_s$ | 0.0017, [19] | [0.001105, 0.00221], estimated | Uniform |
| $\alpha_r$ | 0.0017, [19] | [0.001105, 0.00221], estimated | Triangular, peak 0.0017 |
| $\alpha$ | 0.3, assumed | [0.195, 0.39], estimated | Uniform |
| $r_r$ | 0.0078, [18] | [0.00507, 0.01014], estimated | Triangular, peak 0.0078 |
| $r_s$ | 0.0078, [18] | [0.00507, 0.01014], estimated | Uniform |
| $\beta$ | 0.7, [19] | [0.455, 0.91], estimated | Uniform |
| $\delta_h$ | $1 \times 10^{-3}$, [58] | [0.00065, 0.0013], estimated | Uniform |
| $\mu_{h2}$ | $1 \times 10^{-7}$, [24] | $[6.5, 13] \times 10^{-8}$, estimated | Uniform |
| $\mu_h$ | $4.212 \times 10^{-5}$, [24] | $[2.74, 5.48] \times 10^{-5}$, estimated | Uniform |
| $\mu_v$ | 0.1429, [24] | [0.092885, 0.18577], estimated | Uniform |
| $\mu_{v2}$ | $2.28 \times 10^{-4}$, [24] | $[1.48, 2.96] \times 10^{-4}$, estimated | Uniform |
| $\mu_{v3}$ | 0.0995, [38] | [0.064675, 0.12935], estimated | Uniform |

We then assigned probability distributions to each of the parameters as uniform or triangular, in line with the suggestion of [62]. We used Latin hypercube sampling (LHS) to draw 1000 samples for each of the parameters resulting in a 1000 by 20 matrix, where each row defines a unique parameter set. The parameter sets were used to calculate the reproduction numbers, and then, the partial rank correlation coefficient (PRCC) was used to characterize the statistical contribution of each parameter to the reproduction numbers. The tornado plot of the results is depicted in Figures 3–5.

The top five most sensitive parameters affecting $R_r$ were $\beta_{min}$, $\gamma$, $b_1$, $b_2$, and $\beta$, in that order, as shown in Figure 3. To reduce the value of $R_r$, we need to reduce $\beta_{min}$, $b_1$, and $b_2$ or increase the values of $\gamma$ and $\beta$. It is worth noting that the simultaneous decrease of the value of the parameters with positive PRCC values together with increasing the values of parameters with negative PRCC values will produce a faster reduction of the value of $R_r$. One way to reduce the probabilities of transmission ($b_1$ and $b_2$) is to reduce human-mosquito contacts by using ITNs or LLIN. Increasing $\gamma$ is synonymous with making bed-nets more efficient. This may require regular re-treatment and/or replacing bed-nets as soon as the end of their useful duration is reached; while increasing $\beta$ implies that the proportion of susceptible humans with the sensitive strain should be very high. In other words, there is a need to eliminate the resistant strain completely. One way to make bed-nets more efficient might be to move away from conventional ITNs to LLINs. For $R_s$, the top five parameters in terms of sensitivities are $\beta_{min}$, $\gamma$, $b_1$, $b_2$, and $b$, in that order, as shown in Figure 4. Similar observations can be made as in Figure 3. However, the order of importance of the parameters is slightly different. Here, the proportion of ITN usage appears as one of the top five parameters with high PRCC magnitudes. The implication of this is that in order to reduce the reproduction number $R_s$, we need to ensure that the proportion of humans using ITNs should be as high as possible.

The effective reproduction number $R_{eff}$ combines all the parameters that are in both $R_r$ and $R_s$. Therefore, it is important to calculate PRCC values of parameters that constitute $R_{eff}$. The results of this calculation are shown in Figure 5.

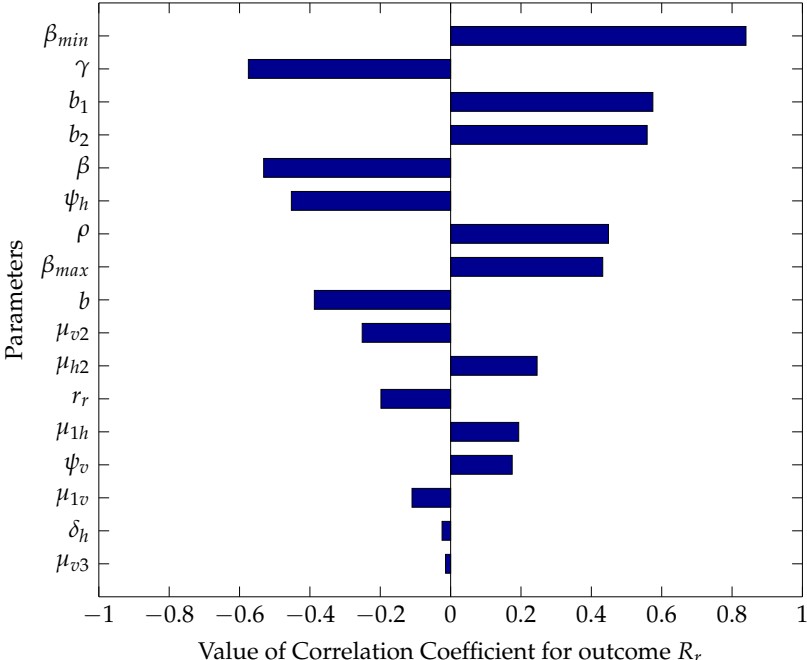

**Figure 3.** Tornado plot showing the sensitivities of the model parameters affecting the reproduction number for resistant strain $R_r$.

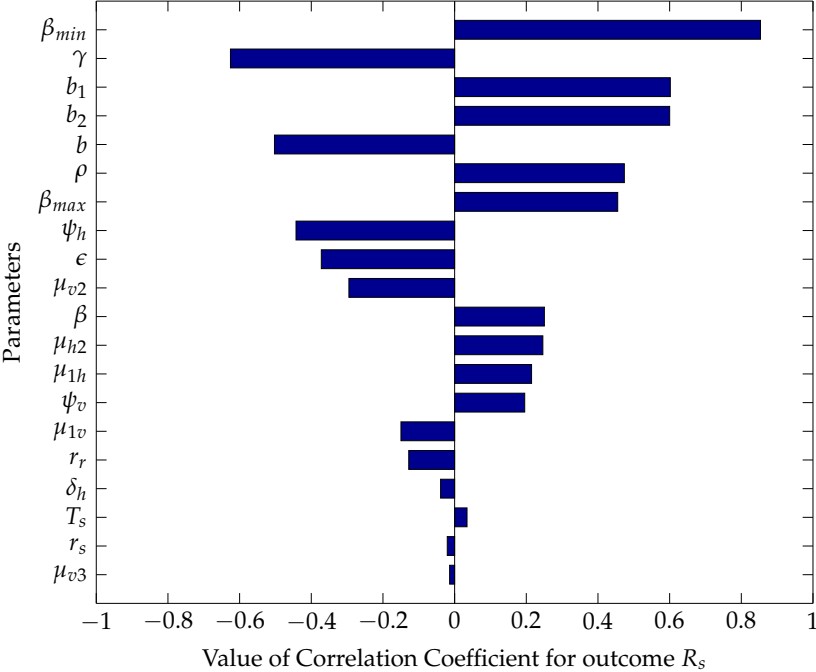

**Figure 4.** Tornado plot showing the sensitivities of the model parameters affecting the reproduction number for sensitive strain $R_s$.

The description of Figure 3 can be extended to Figure 5 with a slight modification. The top five most sensitive parameters affecting $R_{eff}$ are $\beta_{min}$, $\gamma$, $b_2$, $b_1$, and $\psi_h$. Here, $\psi_h$ appears among the top five parameters. Increasing the value of this parameter can lead to a decrease in the value of $R_{eff}$.

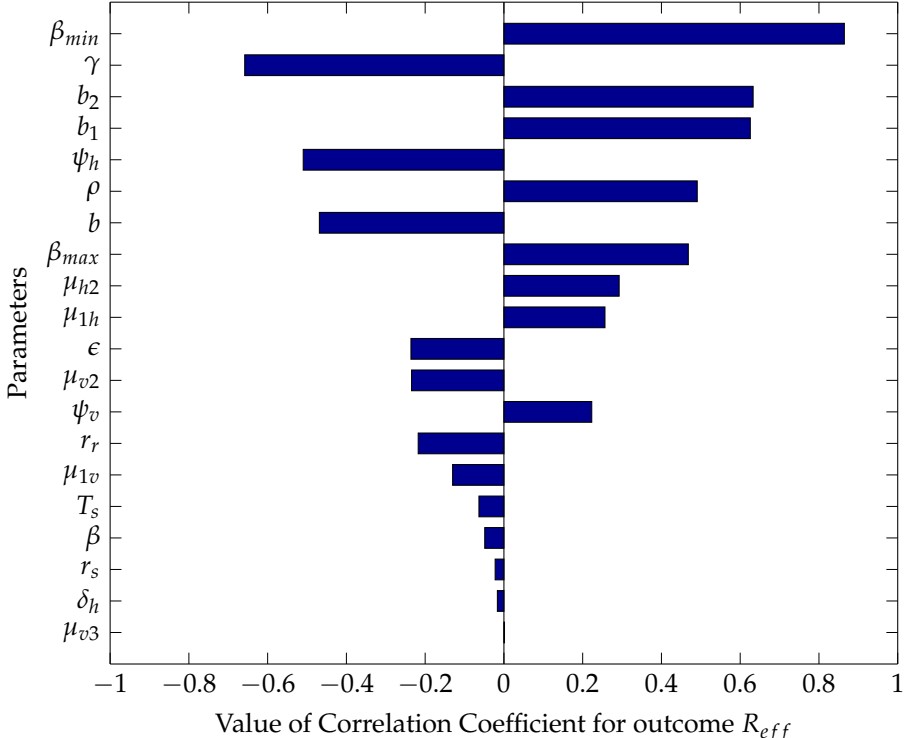

**Figure 5.** Tornado plot showing the sensitivities of the model parameters affecting the effective reproduction number.

Surprisingly, treatment does not appear as part of the top five parameters. This might be attributed to the range of treatment rates used in the study and the fact that only infected humans with sensitive strain are being treated. However, the results further highlighted the importance of ITN and its efficacy towards malaria control. Increasing the efficacy will bring about a reduction in the values of $R_{eff}$, $R_r$, $R_s$, and hence, the scourge of malaria. It is also clear from the result that ITN usage has a bigger impact than treatment in terms of the reduction of malaria spread, meaning that prevention is better than the cure.

### 5.2.2. Numerical Intervention Strategies

By calculating $R_{eff}$ using the 1000 parameter sets discussed in Section 5.2.1, we found that only 210 of these sets gave the value of $R_{eff} < 1$; the remaining were all bigger than one.

The question one may like to ask is: How do we use the intervention parameters $(\epsilon, \gamma, b, T_s)$ to reduce the values of $R_{eff}$? To address this question, we used the 790 parameter sets that gave the values of $R_{eff} > 1$, while keeping the intervention parameters at some predefined values as shown in Table 4.

**Table 4.** Parameter values used for intervention strategies.

| Parameter | $\epsilon$ | $\gamma$ | $b$ | $T_s$ |
|:---:|:---:|:---:|:---:|:---:|
| A | 0.75 | 0.75 | 0.75 | 0.75 |
| B | 0.95 | 0.95 | 0.95 | 0.95 |

For example, $(\epsilon, \gamma, b, T_s) = (0.75, 0.75, 0.75, 0.75)$ is denoted by AAAA, while $(\epsilon, \gamma, b, T_s) = (0.95, 0.95, 0.95, 0.95)$ is denoted by BBBB, $(\epsilon, \gamma, b, T_s) = (0.75, 0.95, 0.75, 0.95)$ is denoted by ABAB, etc. Altogether, 16 different intervention strategies were performed by permuting the values of $\epsilon, \gamma, b, T_s$ using values shown in Table 4. We then calculated the relative effectiveness of a strategy in decreasing reproduction numbers $e_{ff} = \frac{\text{Reproduction number before intervention -Reproduction number after intervention}}{\text{Reproduction number before intervention}} \times 100$. Therefore, if the reproduction number before intervention is very close to the reproduction number after intervention, $e_{ff}$ will be close to zero. The best strategy is one with the highest value of eff. The results of these calculations for $R_s$ and $e_{ff}$ are depicted as box and whiskers plots and presented in Figures 6 and 7, respectively.

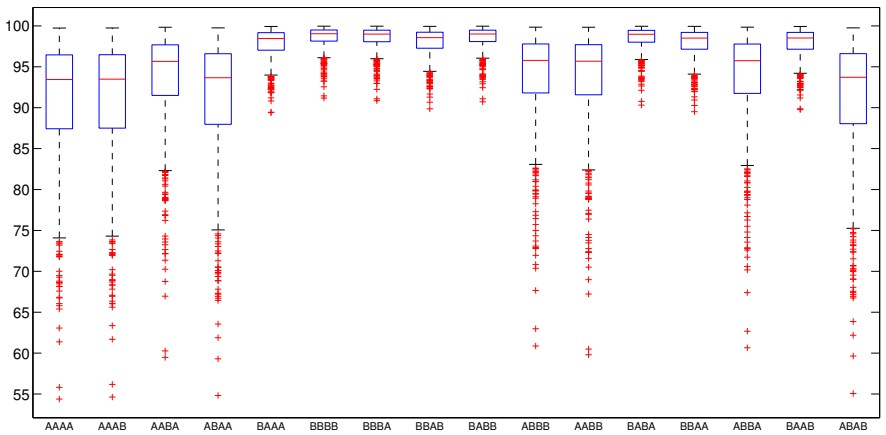

**Figure 6.** Box and whiskers plot showing the results of various intervention strategies on the reproduction number for humans with the sensitive strain $R_s$.

From Figure 6, we observe that for the strategies BBBB, BBBA, and BABB, the boxes shifted up and were smaller compared to the others. These strategies had the effect of reducing the mean values $R_s$ in the range 91–98.6% of the initial values. On the other hand, AAAA and AAAB appeared to have comparatively the least effectiveness. These have larger boxes, larger whiskers, and larger outliers. By comparing BAAA and ABAA, then ABBA and ABAB, one may conclude that the order in which the intervention parameters are taken is important. The reason for this conclusion is that there are three A's and one B in different orders in each one of these. However, BAAA is better than ABAA, which is better than AABA, which is better than AAAB. Furthermore, BAAA is better than ABBB. In other words, it is not just high intervention parameter values in the combinations that matter, the appearance of B in the first position also counts. General observations of Figure 6 indicate that strategies starting with the letter B followed by strings of B's or A's or their combinations are better. The public health implication of these observations is that interventions starting with high treatment efficacy ($\epsilon = 0.95$) should commence first before varying other intervention parameters. Overall, the best strategy is marginally BBBB. This means treatment should be 95%, bed-nets should stay 95% effective, 95% of the population should be using bed-nets, and treating 95% malaria cases. If this strategy can be adopted, malaria can be eradicated. Note that our suggestion for 95% ITN coverage is in line with the WHO's recommendation, whose goal is universal coverage (see [63]).

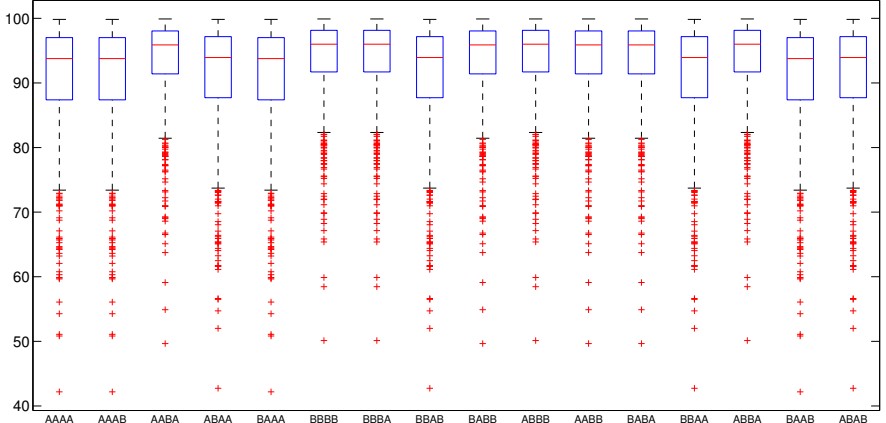

**Figure 7.** Box and whiskers plot showing the results of various intervention strategies on the effective reproduction number $R_{eff}$.

From Figure 7, it is clear that there is no unique best intervention strategy. The best intervention strategy should involve one of the following; BBBB, BBBA, ABBB and ABBA, while AAAA, AAAB, BAAA, and BAAB have the least effectiveness. These strategies have the effects of reducing the mean values $R_{eff}$ in the range 91–93.8% of the initial values. The reason why the interventions are more effective on $R_s$ than $R_{eff}$ could be attributed to the fact that two of the intervention parameters $\epsilon$, $T_s$ only affect $R_s$, but not $R_r$. Generally, it can be observed that all the intervention strategies conducted have positive impacts on reducing the values of the reproduction numbers.

## 6. Discussion

Mathematical models can provide insight into several aspects of the control and spread of malaria in the presence of drug-resistant parasites. In this study, we modeled the transmission and spread of malaria by considering human populations with sensitive and resistant strains of malaria parasites within a community. Our biting rate function indicated that even if the entire population used bed-nets, malaria transmission could only be reduced to a certain minimum value. This is a realistic result because bed-nets are normally used at night, and early mosquitoes bites when people are resting outdoors are reported in some studies such as [64,65]. This finding suggests that outdoor biting has a role in malaria transmission. The question of whether or not scaling removes the possibility of backward bifurcation cannot be answered with certainty. This is because at least two different studies that utilized similar scaling as we have done in this work finished with different outcomes in terms of backward bifurcation. For example, in the work of Ngwa and Shu [27], the model did not exhibit backward bifurcation; however, backward bifurcation was reported in the work of Gimba and Bala [38]. Despite these discrepancies, it is reasonable to state that since there is a possibility of backward bifurcation in Model (3), which no longer exists in Model (15), one can say that scaling removes backward bifurcation in this work. Unlike the work of Tumwiine et al. [19], our model shows that even in the presence of drug resistance, treatment can have a positive impact on the control of malaria. The sensitivity analysis results revealed that if massive interventions strategies through significantly improving treatment efficacy, ITN use, and its efficacy are embarked upon, the effective reproduction number can be reduced to below unity. This implies that malaria can be eradicated. The current study agrees with the work of many authors that reported a decrease in malaria transmissions when ITN use is scaled up (see, for example, [7,66]). However, the observational study in Haiti by Steinhardt et al. [65] indicated that mass distribution of ITNs did

not lead to a reduction in clinical malaria. This might be explained by the fact that ITN possession does not necessarily translate into use. This means that human behavior can affect interventions; see also [57]. The public health implication of the sensitivity analysis conducted is that the parameters with the most significant influence on the reproduction numbers can be targeted in an effort to eradicate malaria from a community. Other parameters in our model that can be targeted for interventions are the vector to human ratio ($\rho_1$) and mosquito death rate. A reduction in these parameters can lead to a reduction in the reproduction numbers. The appearance of ($\rho_1$) as one of the top six most sensitive parameters is quite significant because it supports the work of White et al. reported in [30]. From Figure 6, we can infer that any strategy that starts with high treatment efficacy ($\epsilon = 0.95$) is relatively better than any other one that does not. The implication of this is that, in a region where malaria is endemic, it is better to establish treatment that is 95% effective first, then follow it up with other intervention measures. However, this should not be seen to contradict the results of our PRCC calculations in which we stated that treatment is not in the top five most sensitive parameters. This can be explained by the range of treatment rates used in the sensitivity analysis (<0.455). Since reproduction numbers are measures of the propagation of malaria, the new insight obtained from our sensitivity analysis can help policy makers in designing effective control for malaria transmission and, hence, its eradication. One of the conclusions reported in [29] was that the use of medication accelerates resistance in parasite populations. Our simulation results depicted in Figure 2 support this finding because as the treatment rate was increased from 0.35–0.75, the proportion of infected humans with the resistant strain also increased. Thus, treatment has both desirable and undesirable effects. This might mean that treatment could be described as a double-aged sword in the presence of drug resistance. One of the findings reported in [19] indicated that in the presence of drug resistance, treatment has a negative impact on the reduction of the spread of malaria. On the basis of this finding, what other alternatives are available once malaria cases are established? One suggestion from [29] is that an optimum number of patients to be treated should be found so as to prevent the outbreak of drug resistance. Our findings outlined in Theorem 5 and Corollary 1 provide threshold conditions for efficacy beyond which treatment has a positive impact even in the presence of drug resistance. The important implication of this finding is that it will provide policy makers with clear treatment efficacy to target so as to control malaria. The results from the box plots depicted in Figures 6 and 7 show that malaria can be controlled if bed-net coverage and treatment can be scaled up to about 95%. Unfortunately, this could be quite difficult in many African countries due to poverty, poor public health policy, and poor drug quality (see [29]). From the results of our PRCC calculations, we found that the top five parameters that have the most influence on the disease transmission dynamics are $\beta_{min}$, $\gamma$, $b_1$, $b_2$, and $\psi_h$. These do not fully support the findings of Okunneye and Gumel [67], which showed that the top three parameters of their model were the mosquito carrying capacity, transmission probability per contact for susceptible mosquitoes, and human recovery rate. In our model, the mosquito carrying capacity was made up of many parameters and so did not appear explicit in the PRCC calculations. It is to be understood that any one of the malaria eradication strategies depicted in Figure 7 is effective at reducing the reproduction number drastically. This means that setting ($\epsilon, \gamma, b, T_s$) = (0.75, 0.75, 0.75, 0.75) is also an effective strategy. This is not far away from the finding of Ngonghala et al. [56] that up to 60% ITN coverage might be required to control malaria. High ITN coverage (75% or more) will require mass distribution of nets, which require insecticide treatment every 6–12 months, inline with the recommendations of the World Health Organization [63]. An effective intervention campaign can be successful when ITN coverage through mass distribution is scaled up as reported in [68]. However, embarking on mass distribution is not cost effective, as suggested by [69]. The LLIN that might be more cost effective can form an alternative form of personal protection as opposed to conventional nets.

## 7. Conclusions

In this paper, we modeled the dynamics of a two-strain malaria transmission model by incorporating individuals infected with drug-sensitive and drug-resistant parasites in the human population. Using the next-generation operator, we obtained the associated reproduction number for each strain. We showed that if the sum of reproduction numbers is less than unity, the disease-free equilibrium is globally asymptotically stable. We also showed that in a situation where treatment is 100% effective or is completely absent, the coexistence equilibrium point is globally asymptotically stable where it exists. The global uncertainty and sensitivity analysis conducted showed that if about 95% of malaria cases can be treated with fewer than 5% treatment failure in a population with 95% ITN usage that remains 95% effective, malaria can be controlled. Our analytic intervention calculations on the effective reproduction numbers show that it is more effective to use a combination of strategies in controlling malaria than using only one. We also find that the basic reproduction number is guaranteed to be bigger than the reproduction numbers in the presence of one or two intervention parameters whenever the ratio of the rates at which humans with the sensitive malaria strain acquire immunity to that at which humans with the resistant strain acquire immunity is less than unity. We find that when using a combination of intervention strategies, the order in which the intervention parameters are taken is important.

**Author Contributions:** The conceptualization of the model was done by B.G., which S.I.B. reviewed and approved. B.G. also conducted the model scaling, the positivity analysis, the calculation of the reproduction numbers, and the analytic intervention strategies. S.I.B. constructed the Lyapunov functions and conducted the global sensitivity analysis. The drafting of the manuscript was done by B.G., and S.I.B. substantially revised it. The submission was approved by both authors.

**Conflicts of Interest:** The authors declare that there is no conflict of interest regarding the publication of this paper.

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
