# Peer review of "Global Sensitivity Analysis to Study the Impacts of Bed-Nets, Drug Treatment, and Their Efficacies on a Two-Strain Malaria Model"

_mca, doi:10.3390/mca24010032_

Round 1

Reviewer 1 Report

In page 3 the authors say that the exposed classes were excluded to reduce the number of compartments and make the model analysis easier, maybe a better justification could be given, than making the analysis easier. The justification (2) seams more appropriate.

Page 3, it misses ( ) in " the choice of 1 emanates".

Page 13, line 247, it misses the number of the Figure in "Shown on the Figure are".

Author Response

Response to Reviewer 1 Comments

Point 1: Page 3, it misses ( ) in " the choice of 1 emanates".

Response 1: We have inserted (), it now reads the choice of (1) emanates

Point 2: The justification (2) seams more appropriate.

 Response 2: Thanks, justification (1) has been removed.

Point 3: Page 13, line 247, it misses the number of the Figure in "Shown on the Figure are".

Response 3:  We have corrected that it now reads “Shown on the Figure 2 are”

Reviewer 2 Report

Kindly refer to the attachment.

Author Response

Response to Reviewer 2 Comments

Point 1: The article is marred with many typos and poor grammar. I haven’t corrected all of them. I

suggest that the authors give their work to a colleague and a scientific writing expert to edit it before it can be re-submitted.

Response 1: We have edited with the assistance of colleague. We are now re-submitting.

Point 2: The title suggests that the main aim was to investigate ITNS efficacy but what is written in the abstract is different. i.e. ‘We formulated a deterministic model to investigate the effect of drug resistance on the transmission dynamics of malaria in human population.’ The paragraphs in lines 55-77 also suggest otherwise i.e. ‘The current study extends the work of [10,16] by 1. proposing a simple model of mosquitoes biting rate as a nonlinear function of ITNs usage and included a parameter in the function that represents ITNs efficacy. 2. conduction of global sensitivity analysis to determine the most important parameters governing the dynamics of malaria transmission. This will help in devising optimal intervention strategies that will offer more realistic predictions towards controlling malaria spread. 3. investigating wide range of intervention strategies through sensitivity analysis to determine the most effective strategy for controlling malaria.’ My question is, how is this reflected in the title? It looks like your study investigated the efficacy of several parameters for

example ITNs, drug resistance etc?

 Response 2:  We have done some revisions to the title, the Abstract and the previously lines 55- 57 as follows:

·         New Title: “Global Sensitivity Analysis to Study the Impacts of Bed-nets, Treatment and Their Efficacies On a Two-Strain Malaria Model.”

·         The abstract: Malaria is a deadly infectious disease which is transmitted to humans via the bites of infected mosquitoes.  Antimalarial drug resistance has been identified as one of the characteristics of malaria that complicates control efforts. Typically, the use of insecticide-treated bed-nets (ITNs) and treatment are some of the recommended control strategies against malaria.  Here, the use of ITNs, treatment and their efficacies and evolution of antimalarial drug resistance are considered to be the major driving forces in the dynamics of malaria transmissions. We formulated a deterministic model of two-strain malaria model to investigate the effects of ITNs, treatment and their efficacies on transmission dynamics of malaria in a human population. We propose a simple mosquitoes biting rate function that depends on both proportion of ITN usage and its efficacy. We show that both disease free and co-existence equilibrium points are globally-asymptotically stable where they exist. Global uncertainty and sensitivity analysis conducted show that if about 95\% of malaria cases can treated with fewer than 5\% treatment failure in a population with 95\% ITNs usage that remains 95\% effective, malaria can be controlled. We find that the order in which numerous intervention measures are taken is important.

·         The current study extends the work of [22,37] by

o   Proposing a simple model of mosquitoes biting rate as a non linear function of ITNs usage and included a parameter in the function that represents ITNs efficacy. This will form the basis for studying ITNs usage and its efficacy.

o   2. Investigating wide range of intervention strategies through Global sensitivity analysis to determine the impacts of treatment and its efficacy, ITNs usage and its efficacy in controlling malaria.

o   3. Conduction of global sensitivity analysis to determine the influence of ITNs usage, treatment and their efficacies and other model parameters on the dynamics of malaria transmission. This could help in devising optimal intervention strategies that will offer more realistic predictions towards controlling malaria spread.

Point 3: The authors have obviously not broadly researched on previous work-related to theirs-going

by the small list of references and the tiny introduction. I recommend that they carry out a thorough review of current malaria models related to the investigation of efficacy of treatment options.

Response 3:  We have done more research on previous work particularly on treatment efficacy. We have improved the references substantially. The newly added materials are incorporated in the text.

Point 4: The parameters used in doing numerical simulations are chosen from various sources. I

suggest that a new section is created; only devoted to justifying these parameter values.

Response 4:  We created a new subsection “4.0.1. Baseline Parameter Values and their sources.”  In this subsection, we justified the parameter values used. The new subsection is incorporated in the text.

Point 5: Give a thorough explanation of all your results-especially the Figures.

Response 5: In the revised version of the manuscript, we explained all the results and the Figures. Details are given in the text.

Point 6: There is a need for a broader discussion of the results in comparison to previous studies.

Response 6: We have expanded the discussion of the results, we made comparisons with previous studies, in line with the suggestion of the reviewer.  The details are given the Discussion and some other sections of the text.

Point 7: I like your conclusion-it reads nice BUT it’s not exactly what you have done-or at least you

have not discussed your results well to reflect your conclusion or the abstract.

Response 7: The conclusion is exactly what we have done. For example

o   “We also show that in a situation where treatment is100% effective or is completely absent, the coexistence equilibrium point is globally asymptotically stable where it exist.”   This was discussed under Theorem 4 in Section 3.8.

o   “Global uncertainty and sensitivity analysis conducted show that if about 95% of malaria cases can treated with fewer than 5% treatment failure in a population with 95% ITNs usage that remains 95% effective, malaria can be controlled.”  This was discussed in Section 5.2.2.

o   “Our analytic intervention calculations on the effective reproduction numbers show that it is more effective to use a combination of strategies in controlling malaria than using only one.” This was discussed in Section 5, Theorem 5.

o   “We also find that the basic reproduction number is guaranteed to be bigger than the reproduction numbers in the presence of one or two intervention parameters whenever the ratio of the rates at which humans with sensitive malaria strain acquire immunity to that at which humans with resistance strain acquire immunity is less than unity.” This was discussed under Theorem 5 and Corollary 1.

o   “We find that when using a combination of intervention strategies, the order in which the intervention parameters are taken is important.” This was discussed in section 5.2.2

Minor comments

We have re-written the abstract where we tried to clear some ambiguities, we have corrected the typos and the grammatical errors raised from comments 2 to 26. All corrections are incorporated in the text and we are now re-submitting for e-considerations.

Reviewer 3 Report

The paper deals with a deterministic model to investigate the effect of drug resistance on the
transmission dynamics of malaria in human population. The authors propose a simple mosquitoes biting rate function that depends on both proportion of ITN usage and its efficacy. The analysis of the model show that both disease free and co-existence equilibrium points are globally-asymptotically stable where they exist. They also conduct global uncertainty and sensitivity analysis to show that under certain intervention strategies, malaria can be eliminated and that ITNs efficacy is an important parameter in the reduction of malaria transmissions. They find the threshold condition for basic reproduction number to be bigger than the effective reproduction number, and also find that it is better to establish effective malaria treatment then follow it up with other intervention measures.    

The novelty in the model proposed in this work comes from the formulation of biting rate of female mosquitoes which trnamit malaria transmission. Indeed, unlike in Agusto et al. (2013), they authors propose a non-linear decreasing biting rate ( denoted by $a$) function of proportion of ITNs usage $b$, and its efficacy $\gamma$ (Equation 1). They compute the elasticity index of this new biting rate with respect to $\gamma$ to show that increasing ITN efficacy will decrease the biting rate.

The mathematical results (existence of equilibria and global stability analysis) are well done. But before considering the manuscript for publication, i recommend to the authors to consider these remarks:
    1- For the better illustration of your model, i propose that you draw a compartmental diagram.
    2- In some malaria models which take into account the disease-induced death and non linear force of infection, the phenomenon of backward bifurcation appear. Can you explain clearly in the text why this phenomenon does not appear in yours model? In other words, scaling the population of each class by total species population remove the backward bifurcation?
    3- In page 8, can you explain in the text the description of $\rho$ (see equation 9)?\\

Author Response

Response to Reviewer 3 Comments

Point 1: For the better illustration of your model, i propose that you draw a compartmental diagram.

Response 1: We have drawn the compartmental diagram and it is incorporated in the text.

Point 2: In some malaria models which take into account the disease-induced death and non linear force of infection, the phenomenon of backward bifurcation appear. Can you explain clearly in the text why this phenomenon does not appear in yours model? In other words, scaling the population of each class by total species population remove the backward bifurcation?

 Response 2:  We added another section 3.2. Possibility of Backward Bifurcation where we discussed the possibility of backward bifurcation in the full model. In the Discussion section we concluded that scaling removes backward bifurcation in our model, even though we that there are previous study that perform similar scaling and still backward bifurcation was reported.

Point 3: In page 8, can you explain in the text the description of $\rho$ (see equation 9

Response 3:  We have inserted the description of $ \rho$ in the text as the spectral radius of the next generation matrix.

Round 2

Reviewer 2 Report

The authors have implemented almost all the corrections I suggested. The article reads better with less typographical errors but still with very poor grammar-starting with the title itself. The title is not clear and should further be re-set up.

The title is still not clear (what do the authors mean by treatment?). Malaria treatment involves ITNs, drugs and spraying for example.

The statement in the abstract 'We formulated a deterministic model of two-strain malaria model...' is vague and gramatically wrong. Perhaps it should be re-written to be clear.

I still do not see a sufficient review of previous studies although the introduction is now acceptable.

Author Response

Point 1: The title is still not clear (what do the authors mean by treatment?). Malaria treatment involves ITNs, drugs and spraying for example

 Response 1: Treatment here is restricted to drug treatment. For this reason we modify the title slightly to read “Global Sensitivity Analysis to Study the Impacts of Bed-nets, Drugs Treatment and Their Efficacies On a Two-Strain Malaria Model”

Point 2: The statement in the abstract 'We formulated a deterministic model of two-strain malaria model...' is vague and gramatically wrong. Perhaps it should be re-written to be clear.

Response 2: We have re-written the statement as “We formulate a mathematical model of two-strain malaria to assess the impacts of ITNs, drug treatment and their efficacies on transmission dynamics of the disease in a human population. We propose a simple mosquitoes biting rate function that depends on both proportion of ITN usage and its efficacy.”

Point 3: I still do not see a sufficient review of previous studies although the introduction is now acceptable.

Response 3: We revised the introduction and included newly reviewed materials from literature.